# Ant visual route navigation: How the fine details of behaviour promote successful route performance and convergence

**Amany Azevedo Amin**[ID][1]*, **Andrew Philippides**[ID][1], **Paul Graham**[2]

**1** Sussex AI, School of Engineering and Informatics, University of Sussex, Brighton, United Kingdom,
**2** Sussex Neuroscience, School of Life Sciences, University of Sussex, Brighton, United Kingdom

* aa2645@sussex.ac.uk

**Data availability statement:** Trajectory data is available at https://doi.org/10.25377/sussex.28199639 and associated trajectory

## Abstract

Individually foraging ants use egocentric views as a dominant navigation strategy for learning and retracing routes. Evidence suggests that route retracing can be achieved by algorithms which use views as 'visual compasses', where individuals choose the heading that leads to the most familiar visual scene when compared to route memories. However, such a mechanism does not naturally lead to route approach, and alternative strategies are required to enable convergence when off-route and for correcting on-route divergence. In this work we investigate how behaviour incorporated into visual compass like route learning and recapitulation strategies might enable convergence to a learned route and its destination. Without alterations to the basic form of the initial learning route, the most successful recapitulation method comes from a 'cast and surge' approach, a mechanism seen across arthropods for olfactory navigation. In this strategy casts form a 'zig-zagged' or oscillatory search in space for familiar views, and surges exploit visual familiarity gradients. We also find that performance improves if the learned route consists of an oscillatory motor mechanism with learning gated to occur when the agent approaches the central axis of the oscillation. Furthermore, such oscillations combined with the cast and surge method additively enhance performance, showing that it benefits to incorporate oscillatory behaviour in both learning and recapitulation. As destination reaching is the primary goal of navigation, we show that a suitably sized goal-orientated learning walk might suffice, but that the scale of this is dependent on the degree of divergence, and thus depends on route length and the route learning and recapitulation strategies used. Finally we show that view familiarity can modulate on-the-spot scans performed by an agent, providing a better reflection of ant behaviour. Overall, our results show that the visual compass can provide a basis for robust visual navigation, so long as it is considered holistically with the details of basic motor and sensory-motor patterns of ants undertaking route learning and recapitulation.

analysis code is available at https://github.com/amanyazevedoamin/RecapitulatedTrackConvergence.

**Funding:** AAA was supported by a Leverhulme doctoral scholarship (https://www.leverhulme.ac.uk/). PG & AP were supported by the EPSRC (Engineering and Physical Sciences Research Council, https://www.ukri.org/councils/epsrc/) through the Brains on Board project, Grant Number EP/P006094/1, and the ActiveAI project, Grant Number EP/S030964/1. They were also supported by the BBSRC (Biotechnology and Biological Sciences Research Council, https://www.ukri.org/councils/bbsrc/), Grant Number BB/X01343X/1. The funders had no role in study design, data collection and analysis, decision to publish, or preparation of the manuscript.

**Competing interests:** The authors have declared that no competing interests exist.

## Author summary

Ants are capable of learning long visually guided routes. Given their small brains, it is assumed that they must have a computationally efficient strategy. The current dominant explanation for visual route guidance is that ants choose a direction to move in by comparing the current view (at different orientations) with views stored during learning, a so-called 'visual compass'. These algorithms result in routes which typically run parallel to the training route, and do not explain how insects converge back onto a route when displaced. Using extensive simulations of ant route learning, we explore how innate behaviours and sensory motor strategies during learning and recapitulation can improve performance. We find that a mechanism which integrates a search strategy to sample a wider space by 'casting', interrupted by 'surging' if the visual conditions are favourable, enhances convergence and goal reaching. We also show that altering the basic structure of learning routes to be oscillatory can significantly improve recapitulation. As well as emphasizing the importance of considering behaviour across both learning and recapitulation, this work gives a plausible account of robust route performance that maintains the elegant efficiency expected of insect navigation strategies.

## 1. Introduction

Insects are excellent navigators with many complementary strategies at their disposal [1–5]. These strategies are particularly evident in the striking navigational feats of solitary foraging ants who learn long habitual and idiosyncratic routes [6–8]. With an innate path integration strategy, such foragers explore new terrain and return safely to their nest [9–12]. This is complemented by individuals learning multi-modal sensory information that can be used to navigate in the absence, or even in preference to, path integration [7]. Indeed, for most individually foraging social ants, vision is the dominant sensory modality [13,14]. behavioural evidence suggests that visual information is memorized in an egocentric frame of reference, and that future visual homing or route retracing is carried out by matching currently perceived views to these egocentric visual memories: so-called view based navigation [6,7,10,15–19]. This process is well suited to the small brains and low-resolution visual systems of ants, because it can proceed without the need for complex feature extraction or the explicit identification of landmarks [6,17,20,21].

Guidance using vision is possible, because of the underlying statistics of natural visual scenes. Zeil et al. demonstrated that a memorised view (or snapshot) inherently encodes the orientation and position (i.e. pose) as a result of its egocentric acquisition [20,22]. This is because the difference between a current image and a stored image, as determined by an image difference function (IDF), such as the mean pixelwise difference, increases smoothly between the two images as the agent moves away from the position (translational Image Difference Function, tIDF) or rotates away from the heading (rotational Image Difference Function, rIDF) at which the stored image was acquired. When an agent is within a region where a gradient in the image difference exists, known as the catchment area, the original orientation or position with which a particular snapshot was taken is recoverable through movements which aim to reduce the difference between the current view and the snapshot.

However, visual homing towards a single snapshot does not effortlessly translate to route following. When a chain of snapshots is used to encode a route, route following by homing would require that the agent determines both when a snapshot has been reached and when the next snapshot should be set, a non-trivial task [23]. Baddeley et al. showed that this is overcome if only the rIDF is used for recapitulating the route. An agent placed anywhere

along the route simply needs to perform an on-the-spot scan, comparing the currently perceived view to the set of memorised route snapshots, stepping forward when the current view elicits the most familiarity (i.e when the agent has located the minimum of the rIDF). Repeating this process enables the route to be retraced [23]. These scans are grounded in behaviour, given the on-the-spot rotations ants appear to perform before deciding on their next move [24,25]. Futhermore, evidence from behavioural analysis [19,21,26], agent based modelling [27], robotic implementations [28–32] and computational neuroscience [33,34] all support the idea that ant guidance along these routes can be achieved with this so-called visual compass [20,35], also known as alignment image matching [36], or as in this work, termed view based orientation (VBO) to distinguish from compass mechanisms which rely on celestial visual information. In later work, Baddeley et al. also demonstrated that a familiarity route model could, instead of a bank of snapshots, consist of an artificial neural network (ANN) to provide a holistic representation of the route [27]. Baddeley et al. achieve this with a single hidden layered ANN, trained using the Infomax learning rule. This rule maximizes the mutual information between inputs and outputs, thus performing both a form of feature extraction and memorisation, enabling the encoding of route memories. Other work has also considered how the neural mechanisms for such a representation might be rooted in the mushroom bodies, a pair of insect neuropils with potential for visual associations and therefore a role in route navigation [30,32,33,37] with lesion experiments confirming that these brain structures are essential for visual navigation [38–40].

While view based orientation has become a de-facto insect-inspired navigational algorithm, with implementations on freely moving agents in simulation and robots for routes of up to 60m, across a set of route model types (i.e. snapshot banks, infomax ANN and spiking mushroom body models) [27–30,32,33,41], studies show that when offset from the route but still within a region of familiarity, this strategy generally provides headings that are parallel to the training route, rather than headings which allow for convergence to the route [20,30,41–43]. This is because when comparing views offset a small distance from the route with route views, the best match (i.e. minimum of the rIDF) is an orientation parallel to the route, whereas facing back towards the route (i.e. perpendicular to the learned direction of travel) will never give a good match. Imprecision in movement or sensing resulting in small orientation errors, when followed by instructions which indicate orientations that are parallel, will naturally lead to a drift away from the route as often as a drift towards the route. However, a drift away from the route also compounds the error, as the quality of route matching will then degrade [31], eventually creating a scenario where one final unlucky move puts the agent outside the catchment area of the route views, where it is lost.

Ants do not fail in the same way. Indeed, there is behavioural evidence that individual ants can set directions to converge back onto a route [26,42,44,45]. This may require a reconsideration of whether translational IDF information is also needed for route retracing, as models have demonstrated how this information could be used to home to a specific snapshot location or return towards a route corridor [20,41,46–48]. One common feature between these models is the requirement to 'search' through space for favorable visual familiarity, before moving to exploit this information. Indeed, parallels can be drawn between this and the 'search and surge' mechanisms demonstrated to be conserved across arthropods in olfactory navigation [49]. Here, we wanted to consider if a behavioural search and surge mechanism could be incorporated into view based orientation for route following. As well as considering the recapitulation strategy, we additionally consider whether learning behaviours might shape route memories in ways that promote route convergence.

In this paper, we take a situated approach, wherein agents are embedded in and interact directly with their environment, to address whether view based orientation algorithms are

sufficient for convergent route navigation. This viewpoint emphasizes that view based orientation does not exist in isolation, and that performance may depend on interactions between innate guidance behaviours, motor routines, and the details of the natural world. Overall we show that strategies based on view based orientation can reproduce many of the features of the robust visual navigation of ants, but only when considered in the context of specific adaptive behaviours during learning and recapitulation.

## 2. Models

### 2.1. Route learning heuristics and recapitulation algorithms

To understand how route learning heuristics and route recapitulation algorithms contribute to successful navigation, we test agents using a range of innate behaviours and guidance algorithms. Common to all the experiments is the following: the agent first traverses a training route using innate behaviours (the 'route learning heuristic'). It is during this path that the agent trains an neural network (in this case using the Infomax learning rule) with panoramic views that they experience to provide a holistic encoding of the route which is queried during subsequent route recapitulation. This initial path may be structured to promote view learning which facilitates subsequent route convergence and following. Route recapitulation strategies generally rely on finding the heading at which the currently perceived scene best matches stored route memories. However, there are other ways in which visual matching can influence an agent's movement patterns. A complete strategy is referred to as a 'route learning heuristic + recapitulation strategy'; for example 'baseline + VBO' corresponds to the baseline route learning heuristic and the view based orientation (VBO) recapitulation method.

To assess these strategies for their navigation performance, agents are displaced along a line which runs perpendicular to the training route and passes through its start point (see methods, Fig 6C). Recapitulated paths are then assessed both for convergence back to the training route and also for their ultimate proximity to the route destination (i.e. goal). Before presenting the results, we outline the details of the route learning heuristics and the recapitulation methods that we have investigated, with behavioural strategies represented schematically in Fig 1.

#### 2.1.1. Route learning heuristics

**Baseline route formation: Global vector and obstacle avoidance.**  Path integration is a key navigational strategy for many animals. Whilst visual navigation for route retracing works in the absence of vectors derived from path integration [7,8], it mediates the initial route that is learned [9]. Together with obstacle avoidance, this enables a safe, reasonably-direct route through the environment and is used as the baseline method for route formation [27], a schematic example of which is presented in Fig 1A. During route retracing, where ants can successfully navigate without any information derived from path integration, this global vector is not used but the agent performs the same form of obstacle avoidance as during learning.

**Beacon Aiming.**  As well as general obstacle avoidance, ants are subject to a range of innate biases towards [50] or away from [26,51] certain objects. While insects do not need to extract or label landmarks or complex features for visual navigation [6,17,20,21], conspicuous high contrast parts of a scene are attractive for many foraging ants [50]. It is suggested that attraction to such 'beacons' might allow for a form of route partitioning [50,52]. Behaviourally, beacon aiming can be enacted by biasing routes towards conspicuous objects, as demonstrated schematically in Fig 1B. Alternatively, beacon aiming might be carried out by focusing on a narrower, or restricted, field of view (termed Res. FOV), removing attention from rear or peripheral structures and thus enabling beacons or a cluster of beacons centered

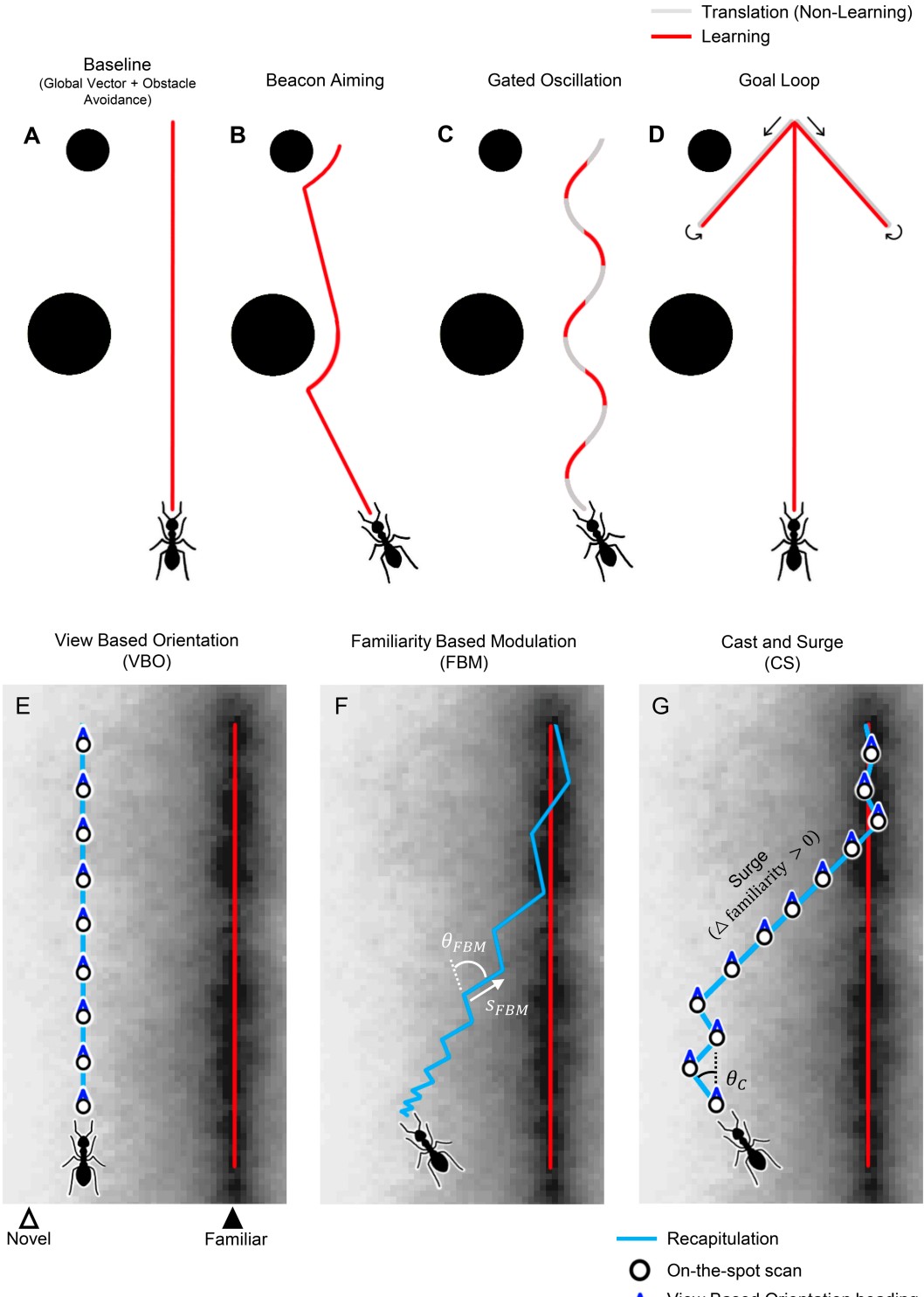

**Fig 1. Schematic illustration of behavioural route learning heuristics (A-D) and recapitulations strategies (E-D), see methods for details of implementation. Route learning heuristics.** (A) Baseline learning route formation uses a global vector together with obstacle avoidance (where necessary). (B) Beacon aiming biases the agent toward salient visual features during learning, with obstacle avoidance taking precedence when within a pre-set proximity threshold (refer to methods). (C) A gated oscillation introduces lateral oscillations into the route, restricting view learning to the portions of the oscillation in which the agent is returning to the central axis i.e. where the phase corresponds to the second and forth quarters of the oscillatory period. (D) An abstract implementation of goal directed learning walks. Additional learning takes place

on path segments to the goal, at 45° relative to the start-goal direction, for the strategy termed 'Goal loop'. **Recapitulation strategies**, where the background represents the relative view familiarity when aligned with the route direction (E) View Based Orientation (VBO) performs an on-the-spot scan to select the most familiar heading at each step. (F) Familiarity Based Modulation (FBM) does not perform on-the-spot scans, instead the agent adjusts turn size ($\theta_{FBM}$) and step size ($s_{FBM}$) based on the familiarity value. (G) Cast and Surge (CS) casts ($\theta_C$) around the heading given by view based orientation, modulated according to the best familiarity from an on-the-spot scan. Casting is suspended in favour of surging if the change in this familiarity is positive. Casting resumes with a small cast angle when the route is reached, essentially reducing the strategy to view based orientation.

within a view to appear more similar from a spread of orientations, perhaps encouraging attraction and therefore route convergence [53]. Here we evaluate, separately and together, behavioural beacon aiming and the restricted field of view for convergence to recapitulated routes.

**Route Oscillations.**   Most ant routes include an oscillation or 'wiggle' about the overall path direction [54–58]. The oscillations are regular and modulated by task, suggesting that they might be active sampling components which enable navigation [59,60]. Oscillations could constitute a route learning heuristic, potentially enhancing future route retracing by widening a region parallel to the overall direction of travel within which familiar views are available. However, if ants learn views evenly spaced throughout the entirety of an oscillatory route, they are subject to nearby views directed both towards and away from the route direction.

One potential solution is to train views in separate memory banks, as shown in previous work where homing views aligned with either the nest or anti-nest direction are assigned to separate banks [61–63], or where memories are divided into left-facing and right-facing banks [59,64]. However, this introduces an added challenge: during the recapitulation phase, a mechanism is needed to determine which memory bank to rely on, or how to combine their familiarity outputs.

The main drawback of this approach is the ambiguity that arises when both networks yield equal familiarity scores, leaving it unclear how the agent should proceed. In the left–right memory bank configuration, such ambiguity can occur when the agent is either facing along the route or directly away from it, two scenarios which should clearly prompt different responses. Similarly, in the nest vs. anti-nest paradigm, ambiguity arises when the agent is oriented perpendicular to the nest–anti-nest axis (i.e. at −90° or +90°). This limitation is apparent in the work of Gattaux et al. [64], who demonstrate that while a bilateral approach can support route following, its effectiveness is restricted to an angular range of approximately ±45° from the route heading.

To address this, we explore a simpler strategy: training a single network solely on views that face back towards the route. During recapitulation, this allows for use of the original view-based recapitulation strategy without any additional steps or modification. To achieve this, we introduce an oscillation and gate learning such that it only occurs on the 'return' portion of each oscillation, i.e. when the agent is moving towards the overall direction of travel, as depicted schematically in Fig 1C. Such gating could promote convergence, as learned views encode some proportion of the vector along the route and the vector facing towards the route. These oscillatory path forms could feasibly be achieved via an intrinsic oscillator, such as the one in the lateral accessory lobes (LAL), which take inputs from a variety of brain regions and outputs to neurons targeting the motor centers [65–67]. The gating of view learning could be controlled by the phase of this underlying oscillator. Indeed the ability to gate by an oscillation's phase has been demonstrated, with wood ants inserting saccade-like turns during

specific, goal-orientated, phases of an oscillatory path to ensure that overall path direction is maintained [55,68].

**Goal-directed learning paths.**  Convergence to a training route may not be essential if the agent still reaches the intended destination. To encourage successful return to a goal (e.g. a fruitful foraging location or a nest) insects undertake learning walks or flights, as demonstrated across ants, bees and wasps [69–73]. During one example of these learning manoeuvrers, ants depart from and repeatedly turn to face the site of interest, the purpose of which is assumed to be the acquisition of views directed towards the target [69]. In simulation, the benefit of these learning paths for accurately reaching a target without overshooting has also been demonstrated [27,61]. Here we implement such learning walks in an abstract way, (Fig 1D), with a simple pair of two-way target-directed learning paths, arranged symmetrically at ±45 degrees to the start - goal direction (i.e. the main route), representing two abstract learning loops where learning occurs only on the return portion of the manoeuvre. We evaluate the performance of these goal directed learning walks, termed 'goal loops' for both route convergence and goal reaching.

### 2.1.2. View based route recapitulation strategies

Within the class of view based navigation algorithms, a variety of strategies have been implemented (e.g. [20,23,27,61,74–76]). However many of these only investigate visual homing to singular goal locations from within the catchment area of a single snapshot, or as captured by multiple goal-directed snapshots, and thus do not consider navigation of the route leading up to the goal and so have not been evaluated for route convergence [20,61,74–76]. Here we evaluate three route following strategies which differ in how they use the information from the rIDF, but which have been proposed specifically for route following.

**VBO: View based orientation.**  The bulk of work implementing simulating insect visual route navigation uses a basic VBO method as the recapitulation strategy [27–29,32,33,77]. Although these models use different approaches to determine the difference in view familiarity between the current view and the stored route views, common to all of them is the mechanism by which the route is recapitulated. To retrace a route, the agent samples a range of orientations by either physical body rotation, or through in-silico rotation of a panoramic view. Then the agent moves in the direction which gave the most familiar view when compared to the route memories, and the process is repeated (Fig 1E). This is equivalent to the agent moving in the direction of the minima of the rIDF at each step. While artificially 'in-silico' rotation is easily implemented, there is uncertainty as to whether insects are capable of this, and it has been suggested that the regular on-the-spot body rotations demonstrated during their routes are a mechanism for rotational sampling [24]. The key point here is that while the value of the IDF is used to determine which is the best heading, the familiarity information is not otherwise used. As all images differences (or familiarities) are assessed from a single location and discarded once the agent takes a step, the agent does not, explicitly or implicitly, access information about the spatial gradient in image difference space (i.e the tIDF).

**FBM: Familiarity based modulation.**  The basic VBO method does not take into account the absolute familiarity value of the views it samples, simply considering which orientation of the current view has maximal familiarity. The method proposed by Kodzhabashev et al. is a klinokinesis inspired approach, where the agent takes the absolute current familiarity and uses it to determine the degree of its next turn and the size of a subsequent step (Fig 1F) [76]. As left and right turns alternate, the agent moves through the environment in an oscillatory fashion. If an agent is faced with low familiarity, this process allows it to explore with large oscillations, gradually reducing the degree of exploration as the familiarity improves.

When the agent is faced with high familiarity, the size of turns reduces towards 0 and the step size is increased, allowing the agent to exploit the favorable familiarity and move directly, enabling route following. As only the current view is used, one of the proposed benefits by the authors of this navigation pattern is that it enables route following without systematic on-the-spot scans [76]. As remarked by the authors, extensive on-the-spot scanning behaviours are typically seen in ants during moments of uncertainty and are not observed during route following [24,76]. In contrast, the established VBO strategy performs on-the-spot scans of a fixed range at every step, determining the rIDF and subsequently setting the heading [27–30,32,33,41]. Therefore a method which relies solely on the familiarity of the current view for route following, and does not perform on-the-spot scans, may be an improved reflection of ant behaviour on familiar routes.

**CS: Cast and surge.**   In the VBO and FBM recapitulation methods discussed above, the strategy remains the same whether an agent's overall direction of travel is up a gradient of familiarity or not. However, when offset from a route, there is always a useful gradient in familiarity space in the direction perpendicular to the route, i.e. back towards the route.

In our third route recapitulation algorithm, 'Cast and Surge', we leverage the information available in the familiarity gradient of the tIDF, alongside the information in the rIDF. The agent first performs a VBO scan, from which it determines the best heading and its associated familiarity score before returning to its original orientation, as this is the direction the agent will surge in if casting does not occur. If the familiarity gradient is unfavorable, as determined by a comparison of this familiarity score to that of the previous step, the directed heading is a sum of the VBO heading and an additional factor. The agent can therefore be thought of as turning to the VBO defined heading, before applying this additional factor. This factor is calculated similarly to the heading in FBM above [76], but is modulated by the familiarity of the view at the VBO determined orientation, rather than the familiarity at the original orientation. It is inversely proportional to the familiarity score, with its polarity changing at each step, allowing for large sweeps, or casts, when in unfamiliar territory, but reducing to allow VBO to dominate when in familiar territory. If the familiarity gradient is favorable, all rotational changes are inhibited and the agent takes a forward step, or surge, at its original heading. Notably, this strategy resembles the cast and surge method conserved across a wide range of arthropods conducting olfactory navigation [49] and has also been implemented in robots for odor following [78]. It therefore seems apt to investigate the efficacy of such a strategy in the visual domain.

In the first part of the results, the full scan range (i.e 360 degrees) is used for the VBO scan at every step. However, we also implement a further step of modulation to reduce this scan range when in an area of high familiarity. At every step, the agent performs a small scan (as implemented by, for example, head rotation) to determine a local familiarity value in the rotational space. The value of the best familiarity encountered in this small scan is then used to determine the size of a subsequent on-the-spot body rotation scan. If the size of this scan is smaller than the initial scan, it is not performed, as the agent is likely within an area of high familiarity and well orientated.

## 3. Results

### 3.1. Cast and surge recapitulation strategies can promote route convergence and goal acquisition

In order to investigate the ability of visual route navigation algorithms to converge to the learned route corridor, we tested combinations of route learning heuristic and route

recapitulation algorithm. To evaluate navigation performance of a test path, we consider two metrics: route convergence and goal reaching. For route convergence, we use ground truth of the training route and measure the mean displacement between the test route and the training route. For goal reaching, we measure the mean displacement between the last 5% of the training route and the test route.

We calculate these metrics for the paths of agents placed at the start of the training route and displaced laterally away from it. For each route navigation method, simulations were run across 75 environment seeds and 45 displacements of up to ±5.5m away from a 20m training route, a schematic of which can be found in the methods (Fig 6C).

As previously reported, the baseline route heuristic and recapitulation method, 'Baseline + VBO' tends on average to result in paths that are parallel to the route [30,41–43]. This is evident in a typical example of a training route and recapitulations (Fig 3A) and is also apparent when considering convergence to the entire route (Fig 2A) or goal reaching (Fig 2D) where results are on average the same as a trajectory running parallel to the route (solid 'Theoretical Parallel' line in Fig 2). An exception is seen for small displacements, where there is divergence away from the route, however this is also to be expected. When offset from the route, the contribution of motor noise has an equal chance to cause convergence towards the route as it does divergence, but when already very close to the route, the agent, drifting either left or right, is diverging in both cases. For these training route lengths and motor noise parameters where catastrophic divergence does not yet occur, such parallel recapitulation on average may be acceptable, but would require an additional mechanism enabling the agent to converge onto the goal destination.

The next question is whether any of the variants of the route navigation algorithm perform better than VBO. 'Baseline + FBM' is worse than VBO on all measures for all displacements (Fig 2A versus 2B, 2D versus 2E). The method can perform approximate route recapitulation and, for instance, in the zero displacement condition, this method on average diverged from the route by 3.25 m after 20m (Fig 2E), similar to the results presented by Kodzhabashev et al. [76]. However, in general, results are poor with paths that diverge quickly and exhibit the maximum turn sizes associated with unfamiliar views (Fig 3B). The issue is that this strategy uses only the familiarity of the current view instead of performing on-the-spot scans. In such a scan, only the orientation parallel to the route and a margin either side will provide familiarity scores which are considered familiar, with the rest of the orientations exhibiting unfamiliar views. An agent faced with an unfamiliar view, and therefore performing large turns and steps, may only find other unfamiliar orientations, as it relies on luck to land within the range of familiar orientations. Therefore there is a risk of catastrophic divergence before being able to find a familiar orientation. As a rotational search is less disruptive to the agent's position than a translational one, it makes sense that an agent would first attempt a methodical on-the-spot scan to anchor itself to a familiar orientation as in the Cast and Surge approach. The original authors note this emergent 'scanning' behaviour [76], but in this work it does not appear to translate to route convergence. As the margin of familiar orientations at any particular position narrows with an increased spatial frequency composition of the environment, it is possible that this method might perform better in environments offering a lower spatial frequency [79].

Cast and surge, as in 'Baseline + CS' is on average better than standard 'Baseline + VBO' across all displacements for both convergence to the route and reaching the destination (Fig 2A versus 2C, 2D versus 2F). To allow statistical comparisons across strategies, we calculate the mean divergence across all starting displacements, such that there is a divergence score associated with each environment seed. This is performed with respect to the whole

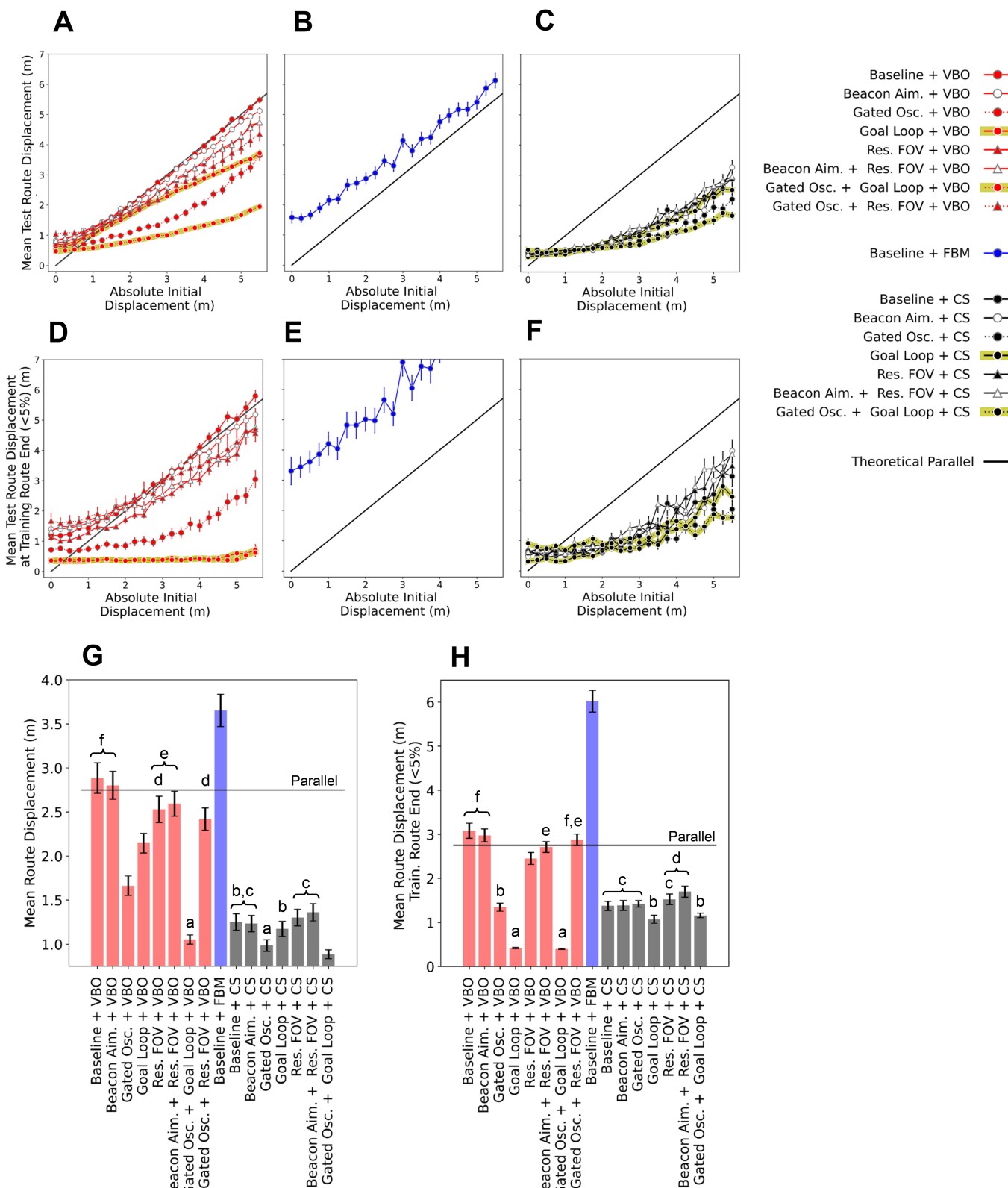

**Fig 2. Divergence pattern and means across strategies as lateral displacement increases. (A-C)** Absolute initial displacements vs mean test route displacements for a 20m learning route, across 75 environment seeds with 45 starting positions (up to ± 5.5m), evaluated with respect to the entire learning route for strategies involving (A) View Based Orientation - VBO, (B) Familiarity Based Modulation - FBM and (C) Cast and Surge - CS. **(D-F)** Similar analysis to the

preceding figures, however now evaluated against the last 5% of the learning route (i.e. for destination reaching). **(G-H)** Mean Test Route Displacements averaged across all initial displacements, again with respect to the entire learning route and last 5% of the learning route respectively. Bars which share a letter indicate groupings which are not statistically significant, as according to a pairwise Games-Howell comparison with $\alpha$ = 0.05. In all cases error bars present standard error on the mean.

training route (Fig 2G) and the final 5% (i.e. goal reaching, Fig 2H), with statistical significance determined by pairwise Games-Howell comparisons ($\alpha$ = 0.05). 'Baseline + CS' is a statistically significant improvement over 'Baseline + VBO' for both route convergence and destination reaching, Indeed Fig 2C and 2F demonstrate convergence on average for all the displacements evaluated, in contrast to the parallel behaviour of VBO in Fig 2A and 2D. The cast and surge method appears less susceptible to loss in comparison with FBM as it engages an initial on-the-spot search to determine the VBO heading, followed by a 'cast' anchored around this heading, thus allowing the spatial familiarity to be sampled around a heading which is generally parallel to the route. This is paired with a mechanism for traveling up positive translational familiarity gradients when the current familiarity is better than the stored value, known as the 'surge'. These surges are particularly evident in the oblique approaches seen in the majority of test routes in Fig 3C, as indicated by the red arrow, which are not unlike the oblique route approaches of ants described by Narendra et al. (Fig 6 in [45]). Fig 3C does also illustrate failures in the 'Baseline + CS' approach, with some test routes on the left hand side of the release points diverging, performing worse than the equivalent 'Baseline + VBO' routes Fig 3A. This demonstrates one of the trade-offs with casting: it might send an agent on the boundary of a catchment area outside it, where there is no useful familiarity information and the agent is essentially lost. This also highlights that the agent is reliant on there always being some useful spatial familiarity gradient.

## 3.2. Route learning heuristics improve route performance and goal acquisition

Given the poor performance of the FBM variant for route convergence, we restricted our analysis of route formation heuristics to the VBO and CS navigation algorithms. Note that the investigated heuristics are in addition to a global vector (as could be derived by path integration) and obstacle avoidance, which form the baseline learning route and is a component of all routes throughout this work. Firstly, we investigated whether modifying the field of view (FOV) during route learning and recapitulation would improve performance as a form of Beacon Aiming. The logic was that reducing the field of view might provide an 'attentional' focus on the parts of the panorama that are aligned with the goal region, potentially offering attraction from displaced locations, a strategy also considered in bio-inspired robotics [53]. We found that reducing the FOV to 252 degrees, (the optimal FOV; for parameter search see supplementary materials: S1 Table and S1 Fig G-H) provided a relatively small but statistically significant improvement over the basic VBO algorithm with full panoramic views, both for route convergence and ultimate destination success (Fig 2G and 2H). Test route examples corresponding to 'Baseline + VBO' and 'Restricted FOV + VBO', as laid out in Fig 3A and 3D respectively, demonstrate this improvement, with the latter showing a slight but distinct narrowing towards the route in comparison to the former. Of note is that in the 'Restricted FOV + VBO' case, paths of the test agents tend to collect beyond the target for most of the training routes evaluated, as can be seen at approximately 5m from the target in Fig 3D. This suggests that reducing the FOV does indeed result in attraction to a single point. However, restricting the FOV with CS did not significantly improve performance over 'baseline + CS' with little

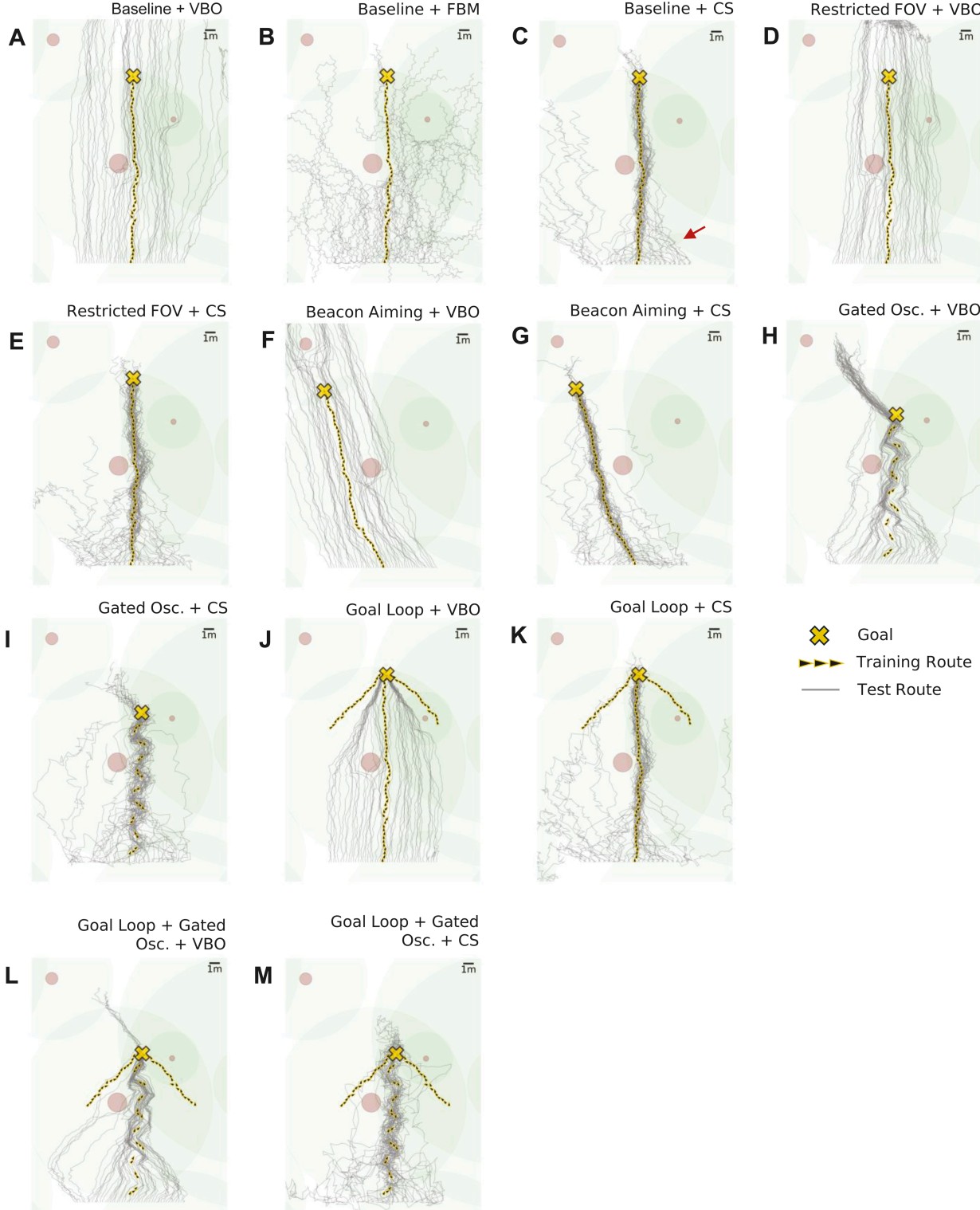

**Fig 3. Training and test route examples across strategies for a single environmental seed**. Yellow corresponds to training route (arrows) and destination (cross), grey corresponds to test routes and trees are illustrated with brown and green circles representing the average trunk and canopy diameters respectively. Presented across strategies of **(A)** 'Baseline + VBO' **(B)** 'Baseline + FBM' **(C)** 'Baseline + CS' **(D)** 'Res. FOV + VBO' **(E)** 'Restricted (Res.) FOV + CS' **(F)** 'Beacon Aiming + VBO' **(G)** 'Beacon Aiming + CS' **(H)** 'Gated Oscillatory (Osc.) + VBO' **(I)** 'Gated Oscillatory (Osc.) + CS' **(J)** 'Goal Loop + VBO' **(K)** 'Goal Loop + CS' **(L)** 'Goal Loop + Gated Osc. + VBO' **(M)** 'Goal Loop + Gated Osc. + CS'.

change in the trajectories, likely because the small change in convergence is not evident when compared to the heading changes brought on by the casts.

Secondly, we examined whether beacon aiming, implemented as attraction to the most prominent object, might improve performance. Examining a typical path shows how beacon aiming during learning causes the agent to shape its route (Fig 3F) such that it heads towards the tallest object in view (in this case, the tree in the top left hand corner). We therefore reasoned that a recapitulating agent might converge at pinch points marked by such beacons. However, we could not see this in the paths, for example, one might expect to see convergence towards the end of the test routes in Fig 2F as they approach the beacon to which the training route aimed for. While it appears to improve VBO slightly at the larger displacements (Fig 2A and 2D), beacon aiming did not significantly improve the baseline versions of either VBO or CS when considered over all displacements, for either route convergence or goal acquisition.

To see if a restricted FOV allowed for more attention on the beacons being aimed for, and thus enable a greater attraction to the beacons, we tested if there was an interaction between the two strategies. However, there was no significant difference in route convergence when compared with just restricting the FOV (Fig 2G) and in fact reduced goal reaching (Fig 2H). Of course, while having little effect on convergence here, beacon aiming might be more beneficial in a sparser environment, where views aimed at beacons from a range of orientations might appear more similar to one another than in this environment, and thus promote convergence. We also note that beacon aiming in ants might not directly enable convergence, but provide other benefits, perhaps ensuring the agent remains in an area which provides a rich visual scene. Alternatively, as beacon aiming is thought to produce distinct route segments, it may be that the routes evaluated are not long enough to consist of multiple prominent beacons and their associated route segments, where perhaps improved performance would arise.

In the tests above, the algorithms utilise learned route views that are aligned with the overall direction of travel. We next explored whether variability in view direction of training views could improve performance. For both VBO and CS, route convergence is significantly improved over baseline by incorporating a gated oscillation into the underlying learning route, with goal reaching also significantly improved over baseline in the case of VBO as a result of this addition. Crucially, the oscillation gating means that views are only learned during the 'return' portion of the oscillation's period ('Gated Osc'), when the ant is oriented towards the overall route direction (as specified by the global vector). This can be seen by the discrete training route segments (yellow arrows) which are obliquely oriented to the overall route direction in the examples presented by Fig 3H and 3I. The improvement in performance is clear in most displacements for VBO (Fig 2A and 2D) as well as the greater displacements for CS (Fig 2C and 2F), and in the summary provided by Fig 2G and 2H. The reason for the success is intuitive, as learning alternately oblique route portions provides intermediate targets, as well as encapsulating both route directed and route-destination directed components, contrasting to the standard approach of acquiring route destination views only. In both VBO and CS, these learning route oscillations result in an oscillatory pattern in the test routes when close to the training route (Fig 3H and 3I). However, for agents at a greater displacement, the view segments pointing towards the route direction are closest and therefore more familiar than the view segments oriented the opposite way. This draws the agent obliquely towards the route, where once close enough it begins to follow the oscillatory path set by both orientations of the view segments. These learning route oscillations widen the familiar route corridor, and therefore also widen the area from within which useful familiar views as well as useful familiarity gradients are available. This explains the improvement in convergence for agents at the larger displacements for 'Gated Osc. + CS' versus 'Baseline +CS' as seen in Fig 2C and the

examples in Fig 3C and 3I. Finally, we note that the recapitulated routes of 'Gated Osc. + CS' now consist of a superposition of oscillatory behaviours, resulting from the navigation algorithm and training route. Both here and for real ants, this highlights an example of the potential complexity of using trajectories to investigate the navigation strategies at work, given that there may be different causes for each of the frequencies involved in oscillatory behaviour.

### 3.3. Goal directed learning walks improve convergence to the goal

We next evaluated the incorporation of goal directed learning walks, consisting of 2 additional training route sections at ±45 degrees relative to the end of the main training route, as could feasibly be acquired from the goal bound portion of a goal directed learning loop (hence the abbreviation 'goal loop'), an example of which can be seen in Fig 3J. Note that the size of these walks is tuned to the divergence of the VBO method for a 20m learning route (see methods). When added to VBO, this significantly improves convergence to the route over baseline, but does not improve on adding an underlying oscillation ('Gated Osc. + VBO') (Fig 2A and 2G). Looking at an example of the training and test route for 'Goal Loop + VBO', Fig 3J, we see that the parallel routes exhibited for VBO in Fig 3A continue until the catchment area covered by the goal directed sections is encountered, where the test routes are drawn into the target (note that the paths do not have to reach the main learning walk locations). Fig 2D shows that this occurs consistently across all initial displacements considered. It is therefore unsurprising that for goal reaching, 'Goal Loop + VBO' does show a significant improvement over both baseline and the best performing previous VBO method of 'Gated Osc. + VBO' (Fig 2H). To capitalise on the route convergence performance gained with an underlying oscillation, and that offered for destination convergence with goal directed walks, we analyse the performance of 'Gated Osc. + Goal Loop + VBO'. This does not compromise the performance of goal reaching (no significant difference between 'Gated Oscillation + Goal Loop + VBO' and 'Goal Loop + VBO', Fig 2D and 2H), but does significantly improve on convergence to the route over 'Goal Loop + VBO' and is also statistically significantly better than route convergence from incorporating an oscillation alone (Fig 2A and 2G). This is evident in Fig 3L, an example of the training and test routes for 'Gated Oscillation + Goal Loop + VBO', showing that the oscillations draw the agents in all along the route, with the goal loop serving to catch any agents which have not converged but are traveling within the area served by the goal loops.

Adding a goal directed learning loop to the CS method significantly improved goal reaching compared to 'Baseline + CS', but did not offer an improvement on route convergence over baseline. Again, we explored the addition of both route heuristics, implementing 'Gated Osc. + Goal Loop + CS'. This provides the best result for route convergence of all of the methods evaluated, but does not improve destination reaching when compared to 'Goal Loop + CS' alone. Furthermore, statistically, neither of these methods compete on destination reaching for goal loop variants of VBO, which are the best performing strategies on this measure. The reason adding a goal loop to any of the CS variants cannot compete with goal loop variants of VBO is that towards the end of the route, the familiarity landscape begins to worsen as views beyond the goal have not been learned. This reduction in familiarity leads to larger casts and causes 'bouncing' around the destination, which is evident when comparing paths around the goal in Fig 3L and 3M.

Lastly, we explored other combinations of strategies to look for potential additive improvements. Given that 'Restricted (Res.) FOV + VBO' and 'Gated Osc. + VBO' both improve on the baseline, we implemented 'Gated Osc. + Res. FOV + VBO'. For route convergence this

performs statistically equivalently to 'Res. FOV + VBO' (Fig 2G), however for goal reaching this performs statistically worse than either 'Res. FOV + VBO' or 'Gated Osc. + VBO' (Fig 2H), suggesting that these strategies may be in conflict. Introducing oscillatory movements may undermine the benefits of restricting the field of view, as they expose the network to a sequence of similar views from slightly different orientations. Essentially, this compromises the directional consistency, perhaps reducing the effectiveness of restricting the field of view in providing a focused visual cue.

## 3.4. Extending the learned route and the implications for homing strategies

While 'Baseline + VBO' showed parallel route following for a 20m route, we theorise that divergence away from the training route grows with distance travelled and that goal directed learning walks, which are currently tuned to the divergence exhibited by VBO at 20m, would have to be adjusted accordingly. As training route lengths grow, so do the mean route displacement scores (as calculated by formula 20) for agents which diverge at the start of their recapitulation, a behaviour termed here as 'rapid divergence'. Including these agents in any averaged results gives an overall indication of the success of the navigation strategy in place, but does not allow for analysis specifically reflecting the ongoing displacement of agents which have had the opportunity to interact with the route.

To investigate longer training routes, we begin by repeating the simulations above, this time extending the route length to 100m and using the baseline strategies of 'Baseline + VBO', 'Baseline + CS' and the two most promising methods (in absence of 'goal loop'), of 'Gated Osc. + VBO' and 'Gated Osc. + CS'. However, we now isolate routes which rapidly diverge, deemed as such if they do not enter a circle of radius equivalent to half the average distance between the start and the goal, as centered on the goal. We first investigate convergence characteristics of non-rapidly divergent test routes with respect to the initial displacement and distance traveled and in doing so, determine the implications with regard to the size of 'goal loops' for agents which have travelled in the correct direction. We then separately examine the rate of rapid divergence, and how this might correspond to the strategy in use.

As expected, divergence grows with route length and initial displacement from the route. The grids in Fig 4A to 4D display the mean route divergence (greyscale colourbar) of test routes at specified distances along the train route, across all the absolute initial displacements considered, for the 'Baseline + VBO', 'Baseline + CS', 'Gated Osc. + VBO' and 'Gated Osc. + CS' strategies respectively. In each case, parallel performance is outlined in red, marking the distance along the training route where the mean displacement from the route matches the initial displacement. While divergence consistently increases with route length and initial displacement from the route, the rate of increase is muted by incorporating strategies previously demonstrated to reduce divergence. Fig 4E to 4H show examples of these routes for a single environment seed. Fig 4E demonstrates increasing divergence with distance traveled, as expected for 'Baseline + VBO'. There is a clear improvement in convergence in the cases of 'Baseline + CS', 'Gated Osc. + VBO' and 'Gated Osc. + CS' (Fig 4F, 4G and 4H respectively), with many of the routes exhibiting the oblique route bound segments also present in the 20m analysis.

The results across 75 environment seeds are summarised in Fig 4I, with groups which are not statistically significant annotated (determined by a Games-Howell pairwise comparison, $\alpha = 0.05$). 'Baseline + VBO' is the only divergent strategy, with 'Gated Osc. + VBO' and 'Baseline + CS' improving on this, providing statistically equivalent converging strategies. When considering these strategies in more detail by looking to Fig 4B and 4C, 'Gated Osc. + VBO'

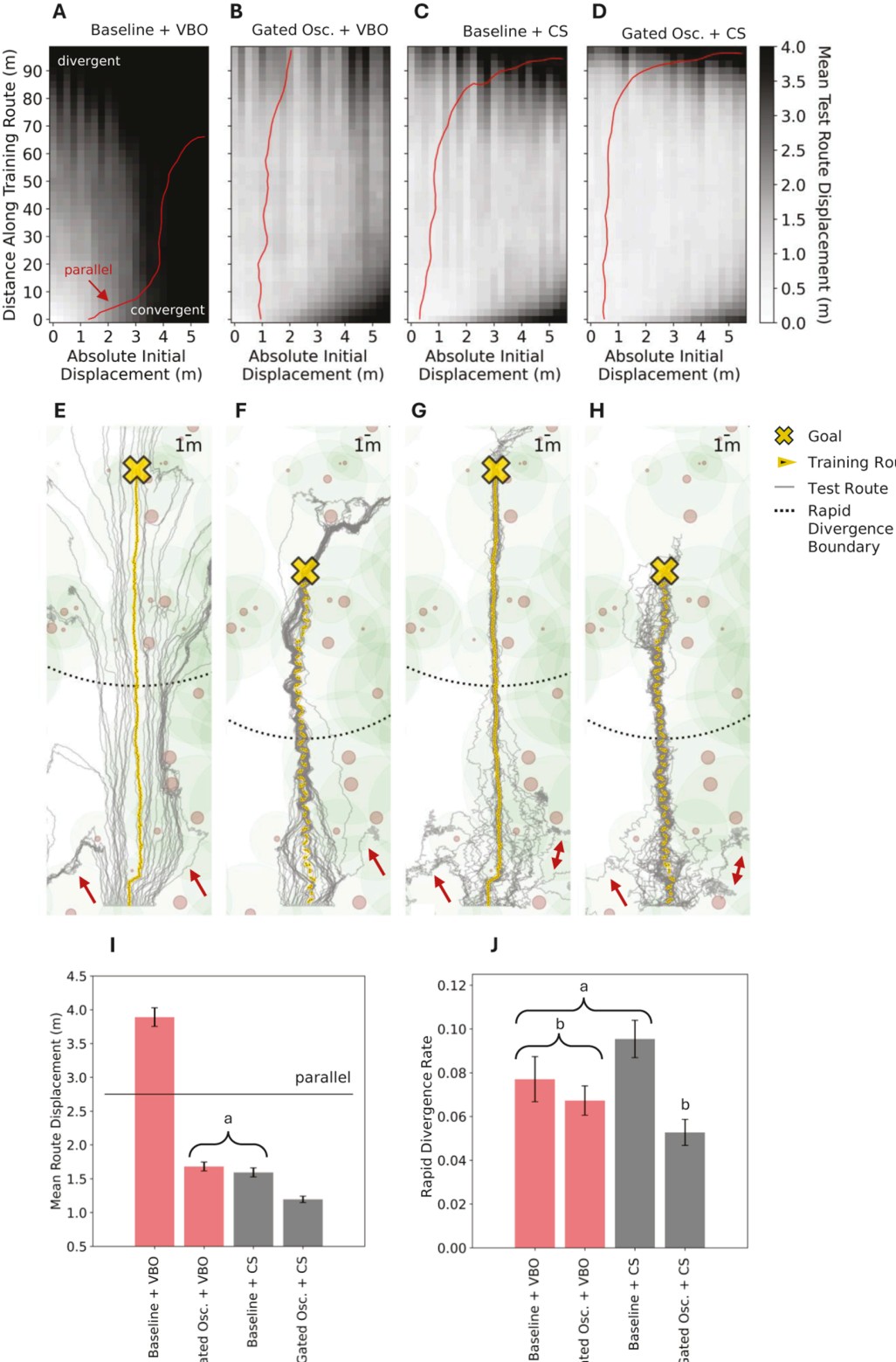

**Fig 4. Divergence depends on length, displacement and navigational strategy. (A-D)** For the strategies of 'Baseline + VBO', 'Oscillatory (Gated) + VBO', 'Baseline + CS', and 'Oscillatory (Gated) + CS': Mean Test Route Displacements as a function of Absolute Initial Displacement and Distance Along Training Route, evaluated across 75 environment seeds. The red line marks the distance along the training route where the mean displacement from the route matches the initial

displacement (i.e. parallel performance). Darker shades are larger displacements. **(E-H)** Example training (100m, yellow) and test (grey) routes across the same strategies, with the cross representing the route end and the rapid divergence radius marked by the dotted black line. Red arrows indicated similar patterns of agent loss **(I)** For these strategies, mean test route displacements for the full route length and across all absolute initial displacements. **(J)** For these strategies, the rapid divergence rate i.e. the proportion of routes which do not enter within a circle of radius equivalent to half the average distance between the start of the training route and the goal, as centered on the goal. For both (I) and (J), all error bars are standard error on the mean and groupings which are not statistically significant are annotated, as calculated according to a Games-Howell pairwise comparison with $\alpha$ = 0.05.

appears to outperform 'Baseline + CS' for convergence at larger initial displacements and route lengths, whereas 'Baseline + CS' outperforms 'Gated Osc. + VBO' for convergence at small initial displacements. This follows if we consider that cast and surge contains a mechanism to tightly follow routes and that the oscillatory learning route widens the area from which useful familiarity information is available. Combining the two as in 'Gated Osc. + CS' provides the best solution of the strategies evaluated in this 100m case, demonstrating both these strengths. However, all these strategies are considered without the 'goal loop' mechanism. As demonstrated in the 20m case, incorporating a pair of 8.2m goal directed learning walks into the VBO strategy provides the best goal reaching overall. As this loop size is determined by the route displacement at the end of the 20m route in the 'baseline + VBO' case, agents that have travelled further and thus diverged further are less likely to encounter the catchment area covered by a loop of this size. Where a goal loop is used for longer routes, it follows that a larger learning loop is required, with the size requirements of this loop being mitigated by use of strategies shown to reduce divergence (i.e. gated oscillation learning routes and/or Cast and Surge).

Finally we consider the rate of rapid divergence (Fig 4J) while 'Gated Osc. + VBO' and 'Baseline + CS' provide a statistically significant improvement in mean route displacement (for non-rapidly divergent routes) when compared with 'Baseline + VBO', the rapid divergence rate between these is not statistically significant (as determined by a Games-Howell pairwise comparison, $\alpha$ = 0.05). Similarly, the rapid divergence rate between 'Baseline + VBO', 'Gated Osc. + VBO' and 'Gated Osc. + VBO' is not significantly different. It therefore appears that there is little influence over these loss rates by the strategy at play, suggesting that these rates might occur at points where useful view familiarity information is scarce in general. Indeed, in the single environment example of route paths presented, Fig 4E to 4H show some common patterns of test route loss (red arrows). The only statistically significant difference is between that of 'Gated Osc. + CS' and 'Baseline + CS', suggesting that incorporating the gated oscillatory training route can mitigate a component of loss brought on by casting.

## 3.5. Scanning can be reduced without compromising performance

As discussed by Kodzhabashev et al, full on-the-spot scans at every step are unrepresentative of ant behaviour, where extensive physical scanning occurs mainly during moments of navigational uncertainty rather than during successful performance of familiar routes [24,76]. In contrast, our implementations of VBO and CS, a full scanning bout of 360 degrees is used at every step. While it could be argued that this full scan could be implemented mentally rather than physically, this doesn't seem consistent with neuroanatomy or behavioural observations [24,55,80,81].

One solution to removing the need for on-the-spot scans is a bilateral approach, where the difference in familiarity of a view as compared to left and right memory banks is used to determine the degree of turning [64,82]. However, these strategies provide similar orientation

commands for agents facing towards or directly away from the route (i.e. where familiarity scores for both memory banks are the same).

Here we propose an alternative approach to reduce on-the-spot scanning, while also allowing an agent to recover when facing any orientation. We investigate whether it is possible to use familiarity to modulate the scanning bout range, as well as the cast angle, without compromising on route following or convergence. For this we use the best performing baseline method, 'Baseline + CS', but in this case, the scan range is modulated by the current familiarity up to a maximum amount (see formula 19), where the current familiarity value is determined in a small 36 degree scan, as could potentially be performed by a small head scan. Performance is evaluated on 5 starting positions up to a lateral displacement of $\pm$2m from the start of the training route. For each set of displacement trials, the maximum scanning range is varied from 36 to 360 degrees. Trials are repeated both with and without modulation of the scan range and 10 environment seeds are considered for each displacement.

As the maximum scanning bout range is increased from 36 degrees to the full 360 degrees, both the modulated and non-modulated cases show a similar pattern consisting of poor initial performance, followed by a decrease in divergence, finally leveling off at a maximum scan range of approximately 160 degrees, from which an increase offers no improvement on convergence (Fig 5A). At this scanning range, there is no statistically significant difference between the mean displacement (across all starting positions and environment seeds) for the modulated and non-modulated methods (Wilcoxon signed-rank test, W = 9, p = 0.06). Therefore, the scanning range can be reduced from the full 360 degrees and modulated without compromising performance. This provides an improved alignment with observed ant behaviour, where 'probability and duration of scanning bouts are increased as a function of how unfamiliar the visual scene is' [24], is consistent with evidence that the number of saccade-like turns reduces with distance to the goal [55] and with recent evidence showing that gaze oscillation frequency reduces on approach to the nest [83]. The advantages of this with respect to time and energy savings are clear. With a maximum scanning bout of 160 degrees, modulating reduces the average number of view interrogations along a route to 75% that of the non-modulated case across the displacements considered (Fig 5B). Such savings are not gained when in less familiar territory where the full scanning range is utilised, but the benefit appears as the agent converges towards the route, as indicated by Fig 5B. Fig 5C and 5D demonstrate this for an example of a single displaced agent recapitulating without and with scanning bout modulation respectively. To begin, both agents are in unfamiliar territory, where they use the full scanning bout range (large dark wedges). As the modulating agent approaches the route in both distance and orientation, the scanning bout range narrows (smaller wedges, light grey). When this agent occasionally veers off course, slightly larger scanning bouts are reinstated. The fixed-scan agent demonstrates similar route convergence, but performs full scans at every step.

## 4. Discussion

Navigation along habitual routes is one of the major insect orientation strategies[6,7,84] and it is assumed to be implemented via alignment image matching [36] also known as the visual compass or view based orientation [20,23,35]. In this work we evaluated route learning heuristics and recapitulation strategies for their ability to produce route and goal convergence, an essential property of a robust navigation algorithm. We show that view based orientation alone is not a convergent strategy (Fig 2A and 2D, Fig 3A), as hinted at in earlier work [20,41,42] and that while recapitulated paths are parallel to the training route at short lengths, they tend to diverge as the training route extends. Complementary strategies are therefore

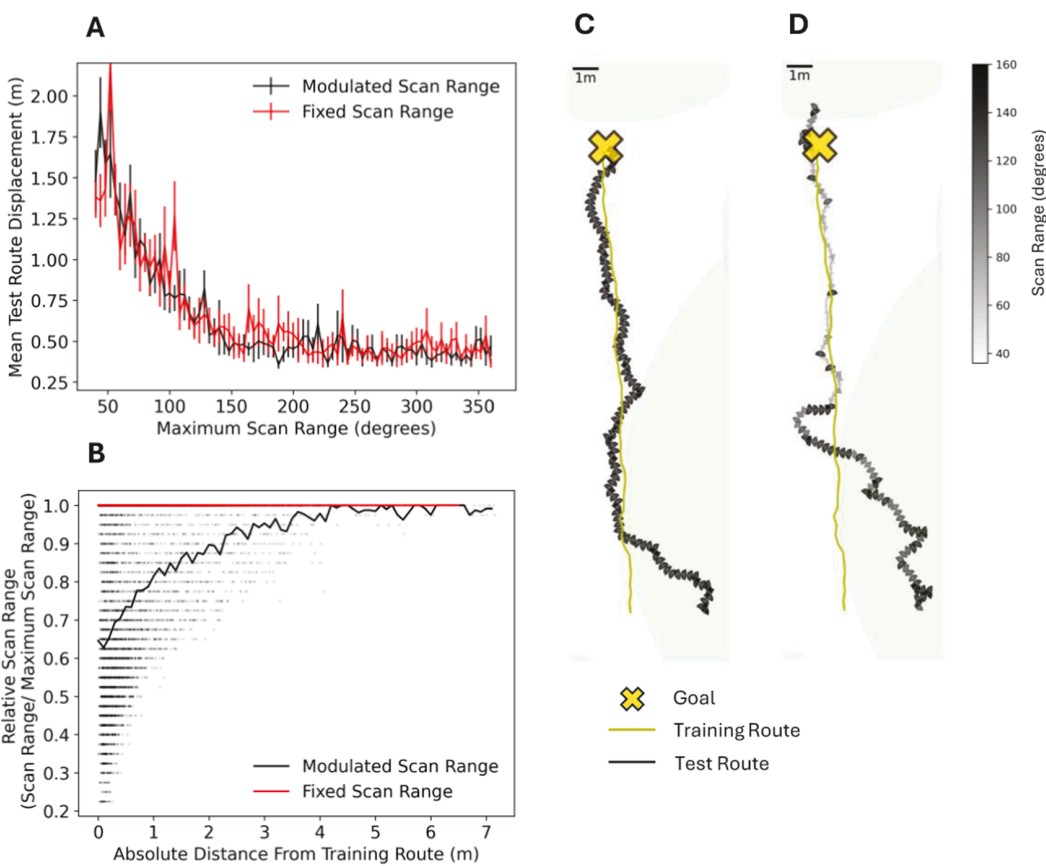

**Fig 5. Scan range can be modulated without compromising performance. (A)** Maximum scan range versus mean test route displacement for baseline + CS with and without modulation of the scan range by familiarity. Evaluated for a 20m training route, 5 displacements from –2m to 2m and over 10 environment seeds. Error bars represent standard error on the mean. **(B)** Distance from Training Route versus individual view scan range relative to the maximum scan range permissible (set at 160 degrees) for Baseline + CS with and without modulation of the scan range. Black markers represent every scan taken by the agent across all 10 environment seeds. Black and red lines represent the mean scan range as a function of absolute distance from the training route for agents operating with modulated and non-modulated scan ranges respectively **(C-D)** Examples of test route (wedges) (C) without and (D) with scan range modulation at 2m displacement for a single seed, converging to a training route (yellow arrows) leading towards a target (yellow cross). Size of wedges and grey colour spectrum represent size of scanning bout.

required to ensure destination reaching, with those incorporating oscillatory mechanisms being particularly effective. More generally, this work highlights the importance of an integrated approach to building and evaluating models of navigation. By examining how recapitulation depends on learning, and how both can be supported by innate motor mechanisms, we emphasise that navigation strategies have been shaped by the co-evolution of brains, bodies, and behaviours within specific ecological contexts and therefore cannot be evaluated in isolation.

Here we begin by considering how a 'Cast and Surge' mechanism, already demonstrated to be widespread in olfactory navigation [49], is capable of both convergence to a route and route following. We then discuss how route convergence is enabled through the incorporation of oscillatory motor and sensor-motor mechanisms in both the learning and recapitulation phases, and that this is consistent with observed oscillatory ant paths. We also discuss

how modulation by familiarity need not just be applied to enable the oscillatory and translatory search mechanism seen in cast and surge, but can also be applied to the on-the-spot scans carried out during recapitulation, and that doing so provides a better reflection of ant behaviour. Finally we consider the difficulty of navigating long routes, the implications this has on homing towards the goal and the increased necessity of employing multiple strategies for successful navigation.

## 4.1. Cast and surge: A single recapitulation mechanism for route attraction and following

We showed that for both route convergence and goal reaching, the cast and surge method improved significantly on the VBO algorithm for recapitulating learning routes formed by the baseline route learning heuristic, offering the desired convergence as opposed to simply parallel route behaviour. This suggests that the 'search and surge' mechanisms conserved across arthropods in olfactory navigation [49] could also serve in visual navigation, or indeed navigation using alternative sensory input types. Given that the mushroom bodies may be responsible for both visual and olfactory memory and associations, as indicated by evidence from neuroscience and behaviour [40,85–87], and also theoretically through computational modelling [33], it perhaps follows that similar recapitulation mechanisms would be used for navigation in both domains. There may already be behavioural indications of search and surge in visual familiarity space. Narendra et al. demonstrated that zero-vector *Melophorus bagoti* ants converge back to a learned 20m route when displaced 10m west of the starting position by initiating a search mechanism, followed by an oblique approach to the route, where, upon reaching it, they engage in visual route following (Fig 6 in [45]). Similar results of this search and approach in zero-vector ants are also demonstrated by Kohler et al. (Fig 10 in [7]). In both cases, it would be difficult to explain behaviour with VBO alone.

Previous work has suggested flipping between distinct mechanisms of route attraction and route alignment as required [36,41,42]. In modeling work, Sun et al. demonstrated a context (i.e. visual familiarity) dependent switch between separate strategies of 'visual homing' for route attraction and visual based orientation for route following. In their homing strategy, the agent is instructed to turn left by an amount modulated by the familiarity of the currently perceived view, as determined by a rotationally invariant encoding based on the frequency content of views. Although this demonstrated successful route approach in the two cases presented (Fig 3D and Fig 5C in [41]), a comprehensive analysis for route convergence was not performed and this strategy may not be well suited to convergence in general. Modulating only a left-ward turn means that sources of noise which send an on-route agent slightly off-route towards the left-hand side are not efficiently counteracted [41]. Secondly, engaging on or off route strategies required the setting of a familiarity threshold, the tuning of which might be variable depending on the environment. Recent implementations on board robots show the success of visual homing by ascending the translatory familiarity gradient, along with odometry, for converging to and following a route [88]. However, again, the issue of determining when the current snapshot has been reached and the next should be set as the target, is shown to be non-trivial and indeed remains unsolved, meaning that this implementation would require parameters to be retuned individually for all routes or environments.

In the cast and surge approach here, navigation does not necessarily require the transition to an alternative mechanism, instead a single strategy is modulated according to the current familiarity and the gradient of familiarity as determined by a one-step memory. When off-route, the familiarity is low, translating to large sweeping exploratory casts being performed *around* the heading given by view based orientation, rather than around that of the current

orientation as in [76]. If the gradient of familiarity is favourable, this ceases all turning or alignment (even towards the familiar orientation), commanding only the original forward direction and therefore in theory allowing for a perpendicular component of travel towards the route. When on-route the casts are down-modulated, to as low as 0 degrees if the familiarity is high, such that the method reduces to view based orientation. Not only does cast and surge present a single strategy regardless of orientation or displacement relative to the route, but it is also robust to a wide range of headings and step sizes (see supplementary materials, S1 Fig), thus it makes an ideal candidate for the next stage of insect inspired biorobotic view based orientation algorithms [28–32].

Modulation of the navigation strategy also need not just be governed by the familiarity triggered by a single network encoding the route. While we found that the presence of a goal directed learning loop improved goal reaching in the alignment method of view based orientation, adding one to the cast and surge method did not attain the same level of performance, where casts increased around the goal location. While this perhaps reflects experimental observations of increased oscillations upon reaching the nest [62,83], we do not rule out the use of multiple memory banks to reduce the need for casts and improve performance. Insects themselves possess multiple mushroom body output neurons, which gives them parallel independent circuits that could potentially encode different categories of view memory [65,89,90]. If insects have independent mushroom body circuits for homing and for routes, then the familiarity of a homing network could inhibit the casting oscillations to a greater degree than that of the route network, essentially converting the cast and surge strategy to basic view based orientation, demonstrated to be successful for the purposes of homing. Additionally, our implementation of homing was simplistic, but it might be improved by combining a pair of networks in opposition as if trained with appetitive and aversive views [61].

Lastly, we note that our implementation of cast and surge is purposefully simple, consisting of a zig-zagging mechanism which changes direction with each step, and a one-step memory comparing the current familiarity with the previous familiarity to determine if surging should occur. It is feasible that an ant might use a broader search strategy, not necessarily consisting solely of oscillations, similar to many of the search strategies seen across olfaction [49]. In these strategies, a search pattern may be paired with an integrator mechanism to continually monitor and update a sensory vector [91–93]. Such an accumulation might be essential in noisier environments, where a simple one-step memory might be too fragile. This is not dissimilar to the mechanism of path integration, in which it is thought that a home vector is stored and updated in the fan-shaped body of the central complex in the form of a sinusoid, the phase and amplitude of which encode direction and intensity respectively [94]. Perhaps this could also serve as the neural basis for accumulating short-term memories [95] of the direction and strength of the gradient of visual familiarity, a form of sensory-based vector [92], with the surge occurring once there is sufficient certainty.

## 4.2. Oscillations as a core mechanism for successful route navigation

One of the most interesting findings from our modelling is that an oscillatory motor pattern is shown to be a useful motif for drawing displaced agents back towards a route. Incorporating a 'gated' oscillation as a training route heuristic enables the best performance for convergence to the route in both the view based orientation and cast and surge cases, with incorporation into the route learning stage of the latter enabling the best performing route convergence strategy overall. Moreover, such an oscillatory mechanism is simple to implement, providing additional opportunities for the improvement of route following in insect inspired robotics. We intuit several reasons for why training route oscillations are helpful. Firstly, they enable a

widening of the area of familiarity parallel to the route. Secondly, views learned now encode both the approach towards the route and towards the goal, to varying degrees depending on where within the oscillation they were acquired. Lastly, as the route consists of distinct segments approaching the central axis of the oscillation for alternate directions, these segments are more distinct from one another than views of similar separation acquired at the same orientation along a route. This may mean that this strategy introduces implicit 'waypoints' but also means that for an offset agent, the closest and therefore most familiar route views are those which have a component directed towards the route, enabling the agents to be drawn towards the route rather than performing parallel following of the route form.

Such a gated oscillation also may obviate the need for a pair of networks encoding opposing directions (i.e. left-hand approach and right-hand approach) [59,60], as it ensures sufficient separation between these banks of views. This is indicated by equivalent performance between single and paired networks (see supplementary materials: S1 Table and S1 Fig), however such antagonistic networks may be necessary to encode left-hand and right-hand route approaches if the translational separation, as offered by the gated oscillation in this case, is not present. The concurrent work of Gattaux et al. [64] and that of Lu et al. [82] indeed train two networks on left and right facing views respectively, as could be gained by an oscillation in the training route, and thus perhaps could provide the same 'drawing in' effect seen in our work if tested for convergence. Indeed, although not systematically evaluated, the strategy of Gattaux et al. shows potential for route convergence (Fig 4B in [64]). Both these works also elegantly demonstrate steering of the agent based on the difference in familiarity of the current view as given by these antagonistic networks, thus removing the need for on-the-spot scans. However, when using such dual networks, an additional mechanism is required to handle the case where both models provide the same familiarity value, as occurs when the signal quality is poor (i.e. with distance away from the route) or in any case where the angle between the current view and each of the network directions is equivalent. This is particularly the case when the current view is oriented directly between the directions represented by each of the networks (somewhat favourable), but is also the case when oriented 180 degrees from this heading (unfavourable). Both represent wildly different orientations with respect to the route and thus should not illicit the same response. Reducing the step size (i.e. speed) as familiarity decreases, as in the work of Gattaux et al, reduces the chance of agent loss by preventing 'running away' when facing unfamiliar directions in general, but does not actually enable any correction. As stated by the authors, their bilateral approach is therefore limited in its range around the route direction in which it can operate. In future work, we would be curious to explore a hybrid strategy between a bilateral approach and some of the suggestions made in this paper. Familiarity-modulated on-the-spot scans would enable an efficient search at various orientations when views are unfamiliar, reducing as familiarity increases. This could direct the agent into the operational range of the bilateral strategy, where no on-the-spot scans are required.

When it comes to the recapitulation phase, oscillations allow for searching and therefore sampling across a wider corridor. From a neuroanatomy perspective, these oscillations which structure learning as well as providing the search mechanism in the 'casts' for recapitulation, may have the same origin in the lateral accessory lobes (LAL). The LAL is a conserved premotor area in insects which takes inputs from a variety of brain regions and whose output are descending neurons targeting the motor centres, thus the LAL have been characterised as a multi-modal coordination centre for steering [67]. The LAL has already been implicated experimentally in 'casting' behaviour in pheromone seeking moths [96,97] and suggested theoretically for oscillatory steering and scanning in ants [57,67,98]. For the involvement of the LAL in a visual cast and surge, it would require input of the visual familiarity for modulating

the cast and that of the rate of familiarity for inhibiting steering (and thus the cast), as could be provided by the mushroom bodies. Indeed, there is evidence of mushroom body output neurons projecting onto the lateral accessory lobe [65,66].

While previous work has investigated using oscillatory casts or zig-zags for homing [61,62] and route following, a thorough displacement analysis for convergence to a route has not been presented [76]. Kodzhabashev et al. theorise that the zig-zagging behaviour replicates the scanning bouts, however we suggest that as the zig-zags or 'casts' are two-dimensional oscillations [54,56–58], they are more similar to the oscillations observed in ant routes [54,56–58], whereas the scanning bouts are distinct fixed in place rotations [24,25]. Furthermore the modeling work of [57] demonstrates routes that consist of a main oscillation, with scanning bouts at the turning points of these oscillations, which is more similar in structure to this cast & surge approach. It also follows that performing an initial orientation scan prior to a translational search is a lower risk strategy, given that there is no commitment to a change in position. While implementing these at every step is perhaps computationally and energetically intensive and unrepresentative of ants, we show that the range of these scans can be successfully modulated by the forward facing familiarity to provide efficient navigation which better matches ant behaviour.

Finally, we remark that a mechanism which results in an increase of the tortuosity of a route might actually enable success, contrary to the classical pursuit of the shortest possible route between two points. Indeed the routes of ants exhibit a tortuosity [9,57,99] which perhaps is not simply attributable to sources of motor noise.

### 4.3. Extending routes and expanding the goal region

Many insects demonstrate highly structured learning behaviours with characteristic loops or turns seemingly designed to increase sensory sampling in and around key locations. Known as learning flights or learning walks, these are theorised to allow for the acquisition of specifically directed learning views enabling future return to a target location [69–71]. Experimentally, Fleischmann et al. showed that limiting the range over which zero-vector *Cataglyphis noda* ants could perform learning walks impacted on both the form of their recapitulated paths and their capacity to return to the nest area [100]. In our work, the interaction of a suitably sized goal directed learning loop with view based orientation is clear, resulting in the best performing set of methods for reaching the goal. However, as the route length or potential displacement from the route increases, so does the risk of divergence on recapitulation, requiring goal loops which may be infeasibly large. For flying insects which are capable of covering large distances with relative ease, the limits of this may reach further than those of walking insects. Indeed, orientation flights, similar to learning flights but spanning several hundred meters, are demonstrated in honeybees [101–104], whereas such walks are demonstrated to be within 10 m for a variety of ant species [69,105].

In the scenario where learning walks are not sufficiently sized for view based route algorithms to succeed without catastrophic divergence, other navigational strategies will be needed. Indeed, as ants often have path integration at their disposal, the size of such learning walks may be preferentially tuned to the drift associated with this mechanism. Narendra et al. demonstrated that zero-vector *Melophorus bagoti* ants exhibit oblique route-directed approaches, as opposed to the nest-directed action exhibited by ants possessing path integration information [44]. However, these ants also perform nest directed learning walks [105], suggesting that knowledge of the nest approach as gained from such a learning walk does not mean that convergent route following strategies are not also employed when using purely view

based navigation. In these scenarios, using additional route learning heuristics and a reca-pitulation strategy such as cast and surge to mitigate the divergence could reduce the size requirements of the goal loop.

The degree to which these strategies are helpful potentially depends on environmental variations, such as depth structure, demonstrated to affect the size of translational catch-ment areas, which in turn impact on the area within which a gradient ascent method such as cast and surge can climb a familiarity gradient [69,79]. In this work we fixed the number of goal directed walks to two and only considered their length, however a myriad of other fea-tures pertain to these goal directed learning walks, such as number of loops and the number and arrangement of smaller scale loops or on-the-spot turns ('voltes' and 'pirouettes') within these. Such variations are demonstrated to be species dependent [73], and therefore might depend on the extent that other route strategies can act given environmental characteristics. All this highlights the need for a holistic approach, also considering the environment in which brains, bodies and behaviours have co-evolved.

## 5. Materials and methods

### 5.1. Simulation environment and agent characteristics

The Unity game engine was used to procedurally generate natural worlds populated with a set density of randomly generated trees as landmarks [106], an example of which is presented in Fig 6A. To ensure that only randomly allocated landmarks were used as cues, a flat and texture-less terrain was populated with eucalyptus trees [107] to beyond the agent's camera

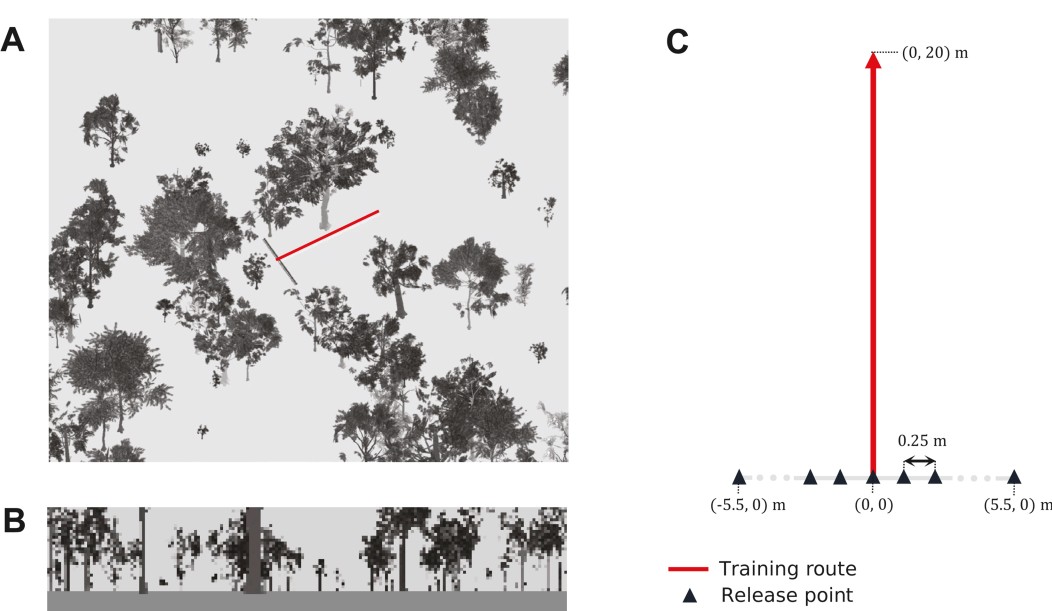

**Fig 6. Simulation environment and experimental procedure**. (A) Example unity simulation environment, with a 20m training route (red) and release points perpendicular to the route and running through the start position (perpendicular black line) (B) Example agent view (as acquired from the start of the route in A), panoramic, greyscale and measuring 36 x 180 pixels. (C) Agents are permitted to carry out a training route according to a route learning heuristic, of length 20m or 100m (depending on the experiment). Agents are then displaced to each of the release points and permitted to recapitulate for a distance 1.5 times the training route length.

clipping distance (set at 50m), with no dominating distal environmental features (e.g. mountains) included. A total of 39 tree models were used, with random selection of location, orientation and size (height range between 0.1 and 30m). The world was populated at a density of 0.007 trees per $m^2$, selected as an approximate replication of the tree feature density present in a Canberra lidar dataset corresponding to the habitat of *Myrmecia croslandi* [108]. Pseudo-random seeds are used so that each combination of route heuristic and recapitulation algorithm can be tested across the same sets of environments. Unless stated otherwise, strategies were evaluated over 75 such environment seeds.

The agent is positioned 1cm off the ground and has a flat elevation (i.e. 0 degrees). In this work, agent views are panoramic and measure 36 x 180 pixels (Fig 6B), corresponding to a resolution of 2 degrees per pixel, selected for consistency with previous work, where this view size was demonstrated to be sufficient for recalling views of virtual routes measuring up to 10km in length [109]. Agents are subject to motor noise drawn for a normal distribution with a standard deviation of 15°, aligning with the work of Baddeley et al. [27].

## 5.2. Route learning

To learn a route an ant first navigates a *training* route, as formed through heuristics (specified below). During this route, views are stored at regular distance intervals and then used to train an artificial neural network (ANN) using an infomax learning rule to encode a holistic representation of the route. On recapitulation, this network is used to output the familiarity of novel views.

We opt for a holistic representation to ensure that the computational cost of model interrogation does not scale with the number of stored route views. This enables the investigation of extended routes without the execution time growing prohibitively large, as would be the case with the 'perfect memory' (also termed 'snapshot bank') approach, in which all route views are stored and each view for recapitulation is compared with every view in storage [23,27]. Although the Infomax approach provides a holistic route representation, it remains an abstraction of the mushroom body. A spiking mushroom body model would offer a closer biological analogue, as it relies on spike-based signalling and local, rather than global, learning. However, simulating spiking dynamics significantly increases execution time while demonstrating comparable performance to Infomax [30]. As infomax has already been shown to perform adequately across the route lengths and view sizes under consideration [109], it remains an appropriate while also efficient choice for exploring the behavioural aspects of route learning and recapitulation.

Training routes are 20m long as derived from experimental work [45] for all results apart from where very long routes of 100m are tested (Sect 3.4). Given the route lengths and view sizes under consideration, images are stored along the agent's path at a density of 5 $m^{-1}$, shown to be sufficient for providing accurate recall when using an infomax-trained ANN [109].

**5.2.1. Encoding routes in an ANN trained with infomax.**  In this work, the ANN and infomax learning rule used to encode the learning route is implemented as per [27,109]. This network consists of a single fully connected layer, the inputs of which are the image pixels, flattened into a one dimensional vector, *s*, of length *N*, from which the activation of the output neurons is calculated as

$$h_i = \sum_{j=1}^{N} w_{ij} s_j \tag{1}$$

where $w_{ij}$ corresponds to the weights between input unit $j$ and output neuron $i$. During learning, outputs are calculated as

$$y_i = tanh(h_i) \tag{2}$$

and weights are updated as follows

$$\triangle w_{ij} = \frac{\eta}{MN}\left(w_{ij} - (y_i + h_i)\sum_{k=1}^{M} h_k w_{kj}\right) \tag{3}$$

$$w_{ij} = w_{ij} + \triangle w_{ij} \tag{4}$$

where $M$ is the number of output units, chosen by default to equal $N/2$ [109], and $\eta$ is the learning rate [110]. Using a uniform random distribution, weights are initialised between –0.5 and 0.5 and standardized to a mean of 0 and standard deviation of 1 [111]. Network weights are updated once for each training view from the training route and then discarded. Following training the novelty of a view is computed as

$$d_{raw}(x) = \sum_{i=1}^{M} |h_i| \tag{5}$$

This learning rule uses mutual information to depress weights associated with the input units (i.e. the view) [112]. Therefore $d_{raw}(x)$ gives an indication of the novelty of a view $x$ when considered relative to the novelty of other views (i.e. the lowest output response is the most familiar view). Another step is therefore taken to normalize the output between a feasible maximum and minimum novelty value.

To determine the minimum possible novelty, a subset of the training images are passed through the trained network for testing. To provide an estimate of a typical novelty output for an unknown image, artificial views with similar statistics to the environment are constructed by scrambling columns of training route views. The minimum novelty (i.e. maximum familiarity) of a sample of training set images, N consisting of images **r** is thus computed as follows:

$$d_{min} = \frac{1}{N}\sum_{i=0}^{i=N} d_{raw}(r_{i=n}) \tag{6}$$

Similarly, the maximum novelty is computed from the training views where the columns are randomly scrambled as follows:

$$d_{max} = \frac{1}{N}\sum_{i=0}^{i=N} d_{raw}(r_{scrambled,i=n}) \tag{7}$$

The normalized novelty of a presented view, $x$, is then computed such that a view which is highly novel has a normalized novelty value approaching 1 and one which is highly familiar has a normalized novelty value approaching 0, via:

$$d(x) = \frac{d_{raw}(x) - d_{min}}{d_{max} - d_{min}} \tag{8}$$

As the infomax network performs a form of feature extraction and memorization, it can be considered an abstraction of the mushroom body, responsible for olfactory and visual associative learning and memory in the brains of many insects [38–40,113–115]. While we could have used a full spiking MB model as in [30], training routes consisted of up to 500 images and across all experiments, billions of model interrogations were performed, thus infomax was the most practical option given time constraints [29,30].

## 5.3. Parameter search

Unless otherwise stated, parameter searches were carried for each strategy described below by setting out a 20m training route, evaluated with 5 test displacements $\pm 2m$ from the starting position, and repeated across 5 novel environment seeds. Parameter search ranges were selected after a rough pass, with the results of this presented in the supplementary materials (S1 Table and S1 Fig).

## 5.4. Route learning and recapitulation heuristics

All learning routes are acquired with the agent moving according to a combination of heuristics. Route recapitulation is also always subject to the heuristic of obstacle avoidance.

**5.4.1. Global vector.** During training, the current orientation is assumed to be given by, at a minimum, a global vector, ensuring directed movement of the agent. Agents are therefore directed from the current position to a distal target coordinate $\vartheta_{target}$. The same target is used for all agents such that all paths head in approximately the same direction, allowing for ease of comparison. All training paths are subject to following this global vector to begin with, with additional route heuristics building on this initial direction. In alignment with Baddeley et al. [27], global vector headings are subject to normally distributed noise with a standard deviation of 5°.

**5.4.2. Obstacle avoidance.** Obstacle avoidance is considered to be a basic requirement of navigation, and is therefore present in learning routes and for recapitulation. Obstacle avoidance is always performed last. During training, the initial directed heading is set according to the learning heuristic or heuristics, whereas during recapitulation, the directed heading is set by the recapitulation strategy. These headings are then adjusted by obstacle avoidance.

Obstacle avoidance first requires extraction of the skyline [35]. Given many insects possess separate UV and green channels, such extraction of the skyline is biologically plausible [116]. If the skyline at the agent's directed orientation is above an (arbritary) threshold of 20 degrees (28% of the view height), the agent turns towards the nearest orientation at which the skyline is below this threshold, that is the nearest edge of the obstacle, thus avoiding the obstacle (in a procedure similar to that used by Baddeley et al. [27]). Otherwise, there is no obstacle to avoid and the agent simply continues with its commanded heading. To ensure only nearby obstacles are avoided, the camera depth for the obstacle avoidance is set to be shallow (arbitrarily as 2m). Ants may use an alternative method (e.g. avoiding the nearest landmark) but our procedure is sufficient as a basic route shaping heuristic.

**5.4.3. Beacon aiming.** We implement beacon aiming in two ways: directly by instructing the agent towards the nearest frontal landmark and indirectly by restricting the field of view.

For the former, we first extract the skyline of the view at full depth, then instruct the agent to orient itself towards the closest part of the skyline to the global vector orientation above a certain threshold (also 10 pixels) within the forward direction of travel, i.e. 45 degrees either side of the directed orientation, $\vartheta_{target}$. As with all other heuristics, beacon aiming is always

followed by obstacle avoidance to ensure orientation towards the edge of a beacon and prevent crashing.

For the latter, we simply reduce the view size by ignoring some of the columns 'behind' the agent. As a default, all views are full panoramas (i.e. spanning 360 degrees) with each column corresponding to 2 degrees for a view size of 36 x 180 pixels. However, focusing on a specific frontal portion of the view might enable learning of beacons without interference from the periphery. Instead of views being made up of $36 \times 180$ rows and columns, the most peripheral 52 columns relative to the heading are dropped, such that the view is now $36 \times 128$ rows and columns, and the field of view restricted to 256 degrees. This FOV was determined by the parameter search (see supplementary materials S1 Table and S1 Fig).

**5.4.4. Oscillatory learning route.** As oscillatory behaviour has been demonstrated in the learning phase of ant walks [57,63,117], the learning route is structured by driving the agent to take an oscillatory path around the global vector heading, $\vartheta_{target}$, by setting the heading via:

$$\vartheta_{\omega}(l_{path}) = \vartheta_{target} + A \times \sin(B \times l_{path}) \tag{9}$$

where $A$ and $B$ are amplitude and period factors, set at 55 degrees and 115 degrees per m respectively and $l_{path}$ corresponds to the distance traveled (in m) from the starting position without oscillations. To ensure that views are only acquired in the portion of the oscillation facing the central axis, view acquisition is gated according to the period such that it is only possible in the second and forth quarters of the cycle. The agent's perception of this oscillation is not subject to any noise, however the spurious view directions learned as a result of motor noise compensate by demonstrating that any methods which rely on this perfect oscillatory knowledge are not dependant on it being exact.

**5.4.5. Goal directed learning walks.** In this work, goal directed learning walks are implemented as a simple pair of target directed learning paths, arranged symmetrically at $\pm45$ degrees to the nest relative to the final nest-ward direction of the training route. For the results in Figs 2 and 3, the length of each goal directed learning walk is set at 8.2m ($=\sqrt{2 \times 5.8 m^2}$), calculated from the maximum divergence of 'Baseline + VBO' (Fig 2E).

## 5.5. Recapitulation strategies

After learning a route, the agent is placed at testing positions spaced every 0.25m up to a distance of $\pm5.5$m either side of the start of the training route along a line perpendicular to the initial vector pointing towards the goal, as per the schematic in Fig 6C. The agents are then instructed to recapitulate using one of the strategies outlined below. Test routes run for 1.5 times the length of the training route before termination (i.e. 30m for a 20m training route), or are terminated if they reach within 0.25m of the goal position.

**5.5.1. View based orientation** The required heading for an agent carrying out View Based Orientation is determined by first calculating the Rotational Familiarity Function (RFF), a variant of the rotational Image Difference Function (rIDF) discussed previously, of the current view, finding its minimum and taking a step, $s_{VBO} = 0.2$m (to match the learning route acquisition density), in that direction:

$$\text{RFF}(x, \phi) = \left[ d(x_{\psi=-\phi/2}), d(x_{-\phi/2+\rho}), ..., d(x_{\phi/2-\rho}) \right] \tag{10}$$

$$\theta_{VBO} = \rho * \underset{\psi}{\text{argmin}}\text{RFF}(x, \phi) - \frac{\phi}{2} \tag{11}$$

The subscript $\psi$ is the orientation of the panoramic view relative to the agent's starting orientation and $\phi$ is the scan range, equivalent to $\phi_{max}$ or 360 degrees in the standard case where the full scan is performed (i.e. equivalent to the agent spinning on the spot and returning to its starting orientation). Therefore where $\psi = 0$, the view is aligned with the starting orientation of the agent. Either by mental or physical rotation, the orientation of the view is rotated (altering $\psi$) to construct a function of heading versus view novelty, the RFF. By aligning to the minimum of this function, the agent can travel in the direction of the most familiar view. As a view $x_\psi$ consists of $n \times m$ rows and columns respectively, the angular resolution, $\rho$ available to a ground based agent in the plane of its motion is as follows:

$$\rho = \frac{\phi_{max}}{m} \tag{12}$$

This standard view based orientation, with a field of view of 360 degrees and m = 180 columns, is the current default for route recapitulation.

**5.5.2. Familiarity based modulation.** The method proposed by Kodzhabashev et al. takes familiarity into account by using it to modulate heading and step size as per equations 13 and 14 [76].

$$\theta_{FBM} = \alpha(-1)^n.d(x_{\psi=0}) \tag{13}$$

$$s_{FBM} = \beta(1 - d(x_{\psi=0})) + s_{min} \tag{14}$$

Where $\alpha$ and $\beta$ are parameters governing the magnitude of the heading and step modulation, set at 80 degrees and 0.15m respectively (see S1 Table), and $n$ corresponds to the number of steps taken, such that through the application of the $(-1)^n$ factor in equation 13, alternate directions are selected, resulting in a zig-zagging pattern. In this work, we also specify a minimum step $s_{min}$, set at 0.45m (determined from the parameter search, see S1 Table and S1 Fig), as without this factor agents often remained stuck on the spot.

Note that in [76], the 'Sum Squared per-pixel Difference' (SSD) is used to determine the familiarity between a current view and a bank of memories representing the route, in a method termed elsewhere as 'Perfect Memory'. In such a case, the value of the lowest novelty is known (i.e. 0), and an indication of the greatest novelty is found by determining the greatest SSD between the first image and the other images in the training set. However, using a bank of views to describe the route is unsustainable for long routes, as memory and computation both grow with route length.

As the infomax method involves holistic training of a neural network with all the training images, it does not suffer in this way. However, using it means that the same method of normalisation cannot be applied, therefore the method set out in 8 is used instead. This uses a subset of the training set passed through the model to determine a minimum novelty score, and a column-scrambled version of the subset to determine an upper bound for the novelty. As infomax is operated within the bounds where it is performant [109], both the infomax and perfect memory methods perform comparably [29,30,33,109] and both methods normalise the novelty of a current view by determining maximum and minimum novelty scores, we do not expect any qualitative differences in agent performance as a result of using infomax instead of perfect memory.

**5.5.3. Cast and surge.** In the default cast and surge method, the RFF over one whole rotation (i.e 360 degrees) is used to initially determine the view based orientation heading and

familiarity. The required 'cast' heading is then computed as follows:

$$\theta_C = \theta_{VBO} + m(x, \phi)\alpha(-1)^n \tag{15}$$

where $m(x, \phi) \in [0, 1]$ is the minimum novelty value of the RFF:

$$m(x, \phi) = \min(RFF(x, \phi)) \tag{16}$$

giving a maximum cast of $\pm\alpha = 60^o$. As all rotation is ceased if a change in novelty is favorable (i.e. negative), the complete 'cast and surge' heading is as follows:

$$\theta_{C\&S} = \begin{cases} \theta_C, & \text{if } \Delta d(x) \geq 0 \\ 0, & \text{otherwise} \end{cases} \tag{17}$$

where $\Delta d(x)$ is the change in the best novelty value (i.e. that which corresponds to the VBO heading) between the current step and the last:

$$\Delta d(x) = m(x_n, \phi) - m(x_{n-1}, \phi) \tag{18}$$

The cast and surge step size, $s_{CS}$ is fixed but depends on the route learning heuristic, as determined by parameter searches (supplementary materials, S1 Table).

**5.5.4. Scan range modulation.**   Thus far, with the exception of FBM, the entirety of the RFF is used. However, this can also be modulated according to the familiarity given by a smaller initial 'head' scan $m(x, \phi_{min})$.

$$\phi_{modulated} = \phi_{min} + m(x_n, \phi_{min}) * (\phi_{max} - \phi_{min}) \tag{19}$$

Where the initial scan range $\phi_{min}$ is set here at 36 degrees. Without modulation, the scan range remains consistent, set to $\phi = \phi_{max} = 360$ degrees unless stated otherwise.

## 5.6. Analysis metrics

**Convergence: route displacement.**   The total displacement of a test route from the training route is calculated as follows:

$$\text{Route Displacement} = \sum_{i=1}^{i=N} \frac{\min_{s \in \mathbf{s}}(D_{L2}(\mathbf{r}[\mathbf{i}], \mathbf{s}))}{N} \tag{20}$$

where N is the number of positions in the train route, $r_i$ is a position vector at index $i$ along the training route, and $\mathbf{s}$ is a the set of vectors representing all positions along the test route. The function $D_{L2}$ calculates the Euclidean distance between the specified index of the train route and each position in the test route. Note that the calculation steps through the train route as opposed to the test route, as otherwise, a test route which stays fixed near the train route would score highly despite making no progress. It also means that routes that pass near the goal but then continue do not count detrimentally to the score.

**Goal convergence: route displacement in the goal region.**   Another metric for determining if a navigational strategy is successful is whether or not the goal has been reached. Some

strategies might enable goal reaching without necessarily attempting convergence, therefore we also consider the route displacement relative to the final 5 % of the training route.

## Supporting information

**S1 Table. Parameter search results across combinations of route learning heuristic and recapitulation method**. Highlighted rows indicate strategies used in the main work. Underline denotes the method to which parameters in the same row correspond. **Notes: (a)** In familiarity based modulation (FBM) no significant difference between having a fixed step or a modulated step (one-way ANOVA, $p = 0.93$). For method consistency with [76] and for computational savings due to a higher minimum step, (a.2) is retained for the rest of this work. **(b)** For cast and surge, a modulated step size is considered, computed as $s_C = \beta(1 - m(x, \phi))$, however, no statistically significant difference was found between having a fixed or modulated step (Wilcoxon signed rank test, $W = 5$, $p = 0.625$), for implementation simplicity (b.2) was retained. **(c)** When the training route is oscillatory and the view acquisitions are gated, there is no statistically significant difference (Wilcoxon signed rank test, $W = 7$, $p = 1$) in performance between using a singular network operating under VBO, or two networks trained on opposing views, for implementation simplicity and computational savings (i.e. one network as opposed to two) method (c.1) is therefore retained. Note that (c.1) has statistically significantly better performance compared to the oscillatory method which does not gate view acquisitions according to the period ($0.61 \pm 0.05$ vs $1.23 \pm 0.05$, Mann–Whitney U = 0, $p = 0.008$). * Parameter derived from [109]. ~ Oscillation Parameters set to those determined from parameter search for 'Oscillatory Gated - VBO'. †minimum step size derived parameter search with fixed step.
(PNG)

**S1 Fig. Parameter search visualisations across combinations of route learning heuristic and recapitulation method, for a subset of methods presented in S1 Table.** Heatmaps present grid searches across two parameters, totally 100 parameter combinations, selected going forward for those which minimised the mean test route displacement, evaluated for (A) Baseline (Obstacle Avoidance, OA) + CS (B) Baseline + FBM, minimum step = 0.45m (C) Oscillatory route (not gated) + VBO (D) Oscillatory route (gated) + VBO (E) Oscillatory route (gated) + Dual Opposing VBO networks and (F) Oscillatory route (gated) + CS. For the baseline VBO method, a parameter search determined that the field of view could be reduced with compromising convergence, but also does not enable it (G) Baseline + Restricted Field of View + VBO (H) Baseline + Beacon Aiming + Restricted Field of View + VBO, error bars represent standard error on the mean.
(PDF)

## Author contributions

**Conceptualization:** Amany Azevedo Amin, Andrew Philippides, Paul Graham.

**Data curation:** Amany Azevedo Amin.

**Formal analysis:** Amany Azevedo Amin.

**Funding acquisition:** Andrew Philippides, Paul Graham.

**Investigation:** Amany Azevedo Amin.

**Methodology:** Amany Azevedo Amin, Andrew Philippides, Paul Graham.

**Project administration:** Amany Azevedo Amin, Andrew Philippides, Paul Graham.

**Software:** Amany Azevedo Amin.

**Supervision:** Andrew Philippides, Paul Graham.

**Validation:** Amany Azevedo Amin.

**Visualization:** Amany Azevedo Amin, Paul Graham.

**Writing – original draft:** Amany Azevedo Amin, Andrew Philippides, Paul Graham.

**Writing – review & editing:** Amany Azevedo Amin, Andrew Philippides, Paul Graham.

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
