## [Decision Letter · Decision Letter 0]

30 Mar 2025

PCOMPBIOL-D-25-00079

Ant visual route navigation: How the fine details of behaviour promote successful route performance and convergence

PLOS Computational Biology

Dear Dr. Azevedo Amin,

Thank you for submitting your manuscript to PLOS Computational Biology. After careful consideration, we feel that it has merit but does not fully meet PLOS Computational Biology's publication criteria as it currently stands. Therefore, we invite you to submit a revised version of the manuscript that addresses the points raised during the review process.

Please submit your revised manuscript within 60 days May 30 2025 11:59PM. If you will need more time than this to complete your revisions, please reply to this message or contact the journal office at ploscompbiol@plos.org. Please include the following items when submitting your revised manuscript:

We look forward to receiving your revised manuscript.

Kind regards,

Julien R SERRES, Ph.D.

Guest Editor

PLOS Computational Biology

Andrea E. Martin

Section Editor

PLOS Computational Biology

**Additional Editor Comments:**

Dear Authors,

All reviewers agree that your study is relevant and significant for publication in the journal PLOS Computational Biology after major or minor modifications.

I encourage you to carefully consider all their comments in order to improve your study.

Thank you for choosing PLOS Computational Biology to publish your research results.

Julien R SERRES, Ph.D.

Guest Editor

PLOS Computational Biology

**Journal Requirements:**

At this stage, the following Authors/Authors require contributions: Amany Azevedo Amin, Andrew Philippides, and Paul Graham. Please ensure that the full contributions of each author are acknowledged in the "Add/Edit/Remove Authors" section of our submission form.

3) We noticed that you used the phrase 'not shown' in the manuscript. We do not allow these references, as the PLOS data access policy requires that all data be either published with the manuscript or made available in a publicly accessible database. Please amend the supplementary material to include the referenced data or remove the references.

5) Thank you for stating that "Trajectory data is available at https://figshare.com/s/0433362aab0b683aadb3". 

We notice that there is a CC BY-NC 4.0 license on your data. We would encourage you to consider using a license that is no more restrictive than CC BY, in line with PLOS’ recommendation on licensing (http://journals.plos.org/plosone/s/licenses-and-copyright). 

6) Please ensure that the funders and grant numbers match between the Financial Disclosure field and the Funding Information tab in your submission form. Note that the funders must be provided in the same order in both places as well. Currently, the order of the funders especially "Leverhulme Trust" is different in both places.

**Reviewers' comments:**

Reviewer's Responses to Questions

**Comments to the Authors:**

**Please note that one of the reviews is uploaded as an attachment.**

Reviewer #1: The authors investigated mechanisms of ant visual route navigation, checked possible strategies required to enable convergence when off-route and for correcting on-route divergence with a situated and embodied approach, to evaluate if view-based orientation algorithms are sufficient for convergent route navigation. The authors discovered that a cast and surge approach, is the most successful recapitulation method, further enhance with an oscillatory motor mechanism with learning, which are verified with agent experiments and systematic comparison. The paper is well written, well structured, with clear introduction, research question, conclusion and insightful discussion, can be published for communication with minor modification, as detailed below.

Minor revision:

1. Verification is almost sufficient to me. Just my curiosity, would it work well, in relatively new environments, e.g. like that in Figure 3 or 4, but test in a similar but different environment? Would be great to show or discuss on this aspect.

2. Figure 1. Caption described the experiments, however, seems not enough to make it clear about the experiment setting and how the experiments have run, add one subplot to describe these schematically may help, or maybe consider swap the order to show those in Figure 2 first in Figure 1.

3. Minor issues:

Line 129, please explain VBO here

Line 149, explain PI when first use, though noticed in line 157

Line 854, weights are updated via, the equation (3) is only the change of weight, how to update weights may also need to explain

Reviewer #2: This is an interesting study investigating different ways in which ants (or agents) could learn routes and retrace them (or not) after being displaced in directions perpendicular to the route. The manuscript needs some work, before publication should be considered.

(1) The Introduction and the first sections of the Results are rambling, need to be shortened and made crisper. My main suggestion is to have a separate section on Models before the Result section and to add a figure in which the different learning and recall regimes that are being proposed and tested in simulation are clearly explained in the form of schematic diagrams.

(2) The Results section contains frequent elements of literature review, discussion and interpretation.

(3) Throughout it remains unclear, how the ‘oscillatory’ models here relate to those proposed by Murray et al. 2020 and LeMoel&Wystrach 2020.

Detailed comments

Line6: Maybe refer also to Huber & Knaden (2015) J Comp Physiol A (2015) 201:609–616 demonstrating very long PI guided trips.

Line8: ‘in preference to PI’: Narendra et al (2013) Proc Roy Soc London B280: 20130683.

Line19: reviewed in Zeil (2023) J Comp Physiol A 209: 499–514.

Line36: ‘the current view’ does not ‘find a minimum of the rIDF’

Line 39: from behavioral analysis?

Line55: Also refer to Hoinville&Wehner (2018) PNAS 115: 2824–2829.

Line59ff: ‘the best match…is an orientation parallel to the route…’. This is certainly true if the rIDF is dominated by distant visual features, but I am not sure whether this is also true in more cluttered environments?

Line69: Also quote Narendra (2007) J Exp Biol 210:1804–1812 here?

Line81: ‘enables future needs for convergence’?

Line82: What is a ‘situated’ approach?

Line86ff: This whole section from ‘Specifically we …’ to ‘…during learning and recapitulation’ should be significantly shortened. It is unnecessary and in places badly worded (see examples below). Without the details of your different model assumptions, it is very confusing.

Line88: ‘exemplars of view based orientation type recapitulation strategies’. Find a nicer way of saying this…

Line90: ‘leads to parallel route following after sideways displacements’?

Line97: ‘what is a learning loop heuristic’? Just try plain language.

Line102: What is VBO?

Line110-line313: This section does not present Results, but a description of Methods and Models. I appreciate that much of it is needed to assess your results, but I suggest to move it to a different heading after the Introduction and before Results, to de-wordify it and to help readers with a figure that explains your different learning and recapitulation schemes with schematic diagrams.

Line119: ‘…paths converge back to the training route after displacements perpendicular to the route’?

Line129: Here’s the VBO again. Unexplained…

Line143-145: Cut.

Line147-line156: Cut.

Line166: What is ‘inherent structure’?

Line192ff: ‘they are subject to …’. Unless they are tagged with the ‘home’ direction, like in the suggestion made by Jayatilaka et al. (2018) J Exp Biol 221: jeb1885306; Murray et al (2020) J Exp Biol 223: jeb210021 and Le Moel & Wystrach (2020) PLoS Comput Biol 16(2): e1007631, that ants may memorize both attractive and repellent views?

Line194: What is a ‘single’ ….model?

Line210: Your references do not cover wasps. If you want to include them, refer to Stuerzl et al (2016) Curr Biol 26: 470-482.

Line211: ‘repeatedly departs from and returns to the site of interest’. That is true for learning walks and the exploration flights of honeybees and bumblebees, but not for the initial part of learning flights.

Line226: IDFs haven’t been defined yet.

Line240: ‘in silico’ rotation.

Line265ff: I do not understand these statements regarding scanning. First, I am not aware of many systematic video-based analyses of scanning along routes and second, why does a ‘method which relies only on the current view’ not require frequent scanning’?

Line274ff: First, how does this ‘Cast and Surge’ strategy relate to LeMoel&Wystrach’s and Murray et al’s algorithm of attractive and repellent views driving the scanning amplitude? And second, does the ‘Cast and Surge’ algorithm explain the observation that ants close to the goal increase scanning amplitude and changes in direction (e.g. Murray et al. (2020) and Zeil (2025) J Exp Biol 2288: jeb249499)?

Line314: This is the first time now that you present ‘Results’. It would be easier for the reader to understand all this if you presented Fig. 2 first, followed by the quantification in Fig. 1.

Line342ff: There are quite a few examples such as this, where you mix methods, results, interpretation & discussion…

Line348: ‘…perform better elsewhere’. What do you mean? On Mars?

Line381: ‘…which form the baseline…’

Line421ff: Again a bit of discussion in the Result section.

Line455ff: What do you mean by ‘potential complexity’? You seem to suggest that it is necessary to also record gaze directions, but a more serious ‘complexity’ may be to distinguish between guidance (how does familiarity drive behaviour) and sampling (the need for scanning)?

Line462ff: with 2J ect are you referring to Fig. 2J?

Line496: ‘…used to train…’

Line544: ‘…results are summarized…’

Line582ff: I don’t quite understand why full 360deg scans would be needed at all, except to get an initial bearing when displaced. In cases where gaze directions of ants on familiar routes have been recorded, there is always an oscillation (e.g. Lent et al (2010) PNAS 107: 16348-16353; Zeil (2025)).

Line589ff: Is this not similar to Murray et al (2020) and LeMoel&Wystrach (2020)?

Line605ff: What is the ‘observed ant behaviour’ you want to ‘align’ with?

Line613ff: ‘…without and with scanning…’. Fig 4C is without and Fig. 4D is with modulation.

Line689ff: Again, is this not very similar what Murray et al (2020) and LeMoel&Wystrach (2020) propose?

Line698ff: ‘…as casts increase around the goal location’. This is exactly what is observed in ants (Murray et al 2020, Zeil (2025) J Exp Biol 228: jeb249499).

Line707: Refer in addition to Murray et al 2020 here.

Line723: Are there good reasons why you don’t refer to Hoinville&Wehner (2018) in this context?

Line796: You may want to refer to more recent work on honeybee/bumblebee exploration flights by Degen et al and Woodgate et al (2016).

Line797: There are data showing that learning walks can be much more extensive: Deeti&Cheng(2021) J Exp Biol 224: jeb242177.

Line811ff: tIDF catchment areas depend on the depth structure of the habitat.

Line812ff: In principle, can ‘cast and surge’ be considered to be a simple (move and compare) gradient descent method?

Line827: It would be really interesting to check the degree to which the absence or presence of distant visual features affects your simulations. The rIDF would be more robust and would provide directional guidance over a larger range, if distant visual features were present.

Line837ff: Views don’t just ‘appear’. Further down it becomes clear that learning is paced to 5 views per metre (of path?), meaning that views are learnt independent of whether they have changed or not. Another interesting future project: views are only learnt, when the scene has changed.

Line842: What does ‘selected for similarity with previous work’ mean?

Line935: Give references for this statement.

Check references. Many are incomplete, species names are not italic ect ect.

Reviewer #3: This study explores an interesting question: how ants can improve convergence to a learned route in navigation, as previous studies have shown that their routes tend to run parallel to the trained path. While I am not a specialist in insect navigation, I was able to follow the core concepts and key contributions of the research. The study is well-structured, clearly presented, and demonstrates the level of thorough investigation expected in high-quality research. From my perspective, I have a few concise and minor concerns that I hope the authors can address in the paper:

1. Centralized vs. Decentralized navigation systems: A debate in insect navigation research concerns whether these systems are centralized or decentralized. As I understand it, different insect species—such as fruit flies (Drosophila), desert ants, and bees—employ varying navigation strategies. Path integration, visual homing, and route following are influenced separately or jointly by multisensory information from subsystems including the global compass, local compass, odometer, and other cues. Given these disparities in navigation mechanisms across species, what are the authors’ perspectives on how this study contributes to the broader understanding of insect navigation? How might the findings be contextualized within existing knowledge of these different species?

2. Potential for Robotic Implementation: How significant is the gap between the observed navigational behavior in ants and its potential implementation in robotic navigation systems? Could the proposed mechanisms be directly applied?

I look forward to the authors's insights on these points.

Reviewer #4: The review is uploaded as an attachment

**Have the authors made all data and (if applicable) computational code underlying the findings in their manuscript fully available?**

Reviewer #1: Yes

Reviewer #2: Yes

Reviewer #3: Yes

Reviewer #4: Yes

PLOS authors have the option to publish the peer review history of their article (what does this mean?). If published, this will include your full peer review and any attached files.

Reviewer #1: No

Reviewer #2: No

Reviewer #3: No

Reviewer #4: No

**Figure resubmission:**
---

## [Decision Letter · Decision Letter 1]

18 Aug 2025

Dear Ms Azevedo Amin,

We are pleased to inform you that your manuscript 'Ant visual route navigation: How the fine details of behaviour promote successful route performance and convergence' has been provisionally accepted for publication in PLOS Computational Biology.

Before your manuscript can be formally accepted, you will need to complete some formatting changes, which you will receive in a follow-up email. A member of our team will be in touch with a set of requests.

Best regards,

Prof. Julien R Serres

Guest Editor

PLOS Computational Biology

Andrea E. Martin

Section Editor

PLOS Computational Biology

Reviewer's Responses to Questions

**Comments to the Authors:**

Reviewer #1: Noticed all my concern and comments have been clearly addressed in the revised version.

Reviewer #3: The authors addressed my concerns. I do not have further comments.

Reviewer #4: Overall, the authors have addressed my concerns to a satisfactory extent.

Reviewer #5: This study presents a test of a cast and surge approach to route following, that appears to outperform simple view-based orientation and familiarity modulation algorithms in simulation.

In my opinion, the authors have done an excellent job at incorporating the proposed changes and literature to place their work in a wider context. I note only a few typographical errors introduced in the process.

Page 6 line 99—“trains a neural network” -> “trains a neural network”

Page 21 line 437—"the size of these walks is tuned” -> “the size of these walks was tuned”

Page 33 line 730—"insect inspired robotics” -> “insect-inspired robotics”

Page 34 line 753—"Both these works” -> “Both of these works”

Page 38 line 859—"Pseudo-random seeds are used” -> “Pseudo-random seeds were used”

**Have the authors made all data and (if applicable) computational code underlying the findings in their manuscript fully available?**

Reviewer #1: None

Reviewer #3: None

Reviewer #4: Yes

Reviewer #5: Yes

PLOS authors have the option to publish the peer review history of their article (what does this mean?). If published, this will include your full peer review and any attached files.

Reviewer #1: No

Reviewer #3: **Yes: **Qinbing Fu

Reviewer #4: No

Reviewer #5: No

---

## [Editor Report · Acceptance letter]

PCOMPBIOL-D-25-00079R1

Ant visual route navigation: How the fine details of behaviour promote successful route performance and convergence

Dear Dr Azevedo Amin,

I am pleased to inform you that your manuscript has been formally accepted for publication in PLOS Computational Biology. Your manuscript is now with our production department and you will be notified of the publication date in due course.

With kind regards,

Zsofia Freund
